# Differentiable Extensions with Rounding Guarantees for Combinatorial Optimization over Permutations

**Robert R. Nerem**
UCSD
rrnerem@ucsd.edu

**Zhishang Luo**
UCSD
zluo@ucsd.edu

**Akbar Rafiey**
NYU
ar9530@nyu.edu

**Yusu Wang**
UCSD
yusuwang@ucsd.edu

## Abstract

Continuously extending combinatorial optimization objectives is a powerful technique commonly applied to the optimization of set functions. However, few such methods exist for extending functions on permutations, despite the fact that many combinatorial optimization problems, such as the quadratic assignment problem (QAP) and the traveling salesperson problem (TSP), are inherently optimization over permutations. We present Birkhoff Extension (BE), an almost-everywhere-differentiable continuous polytime-computable extension of any real-valued function on permutations to doubly stochastic matrices. Key to this construction is our introduction of a continuous variant of the well-known Birkhoff decomposition. Our extension has several nice properties making it appealing for optimization problems. First, BE provides a rounding guarantee, namely any solution to the extension can be efficiently rounded to a permutation without increasing the function value. Furthermore, an approximate solution in the relaxed case will give rise to an approximate solution in the space of permutations. Second, using BE, any real-valued optimization objective on permutations can be extended to an almost-everywhere-differentiable objective function over the space of doubly stochastic matrices. This makes our BE amenable to not only gradient-descent based optimization, but also unsupervised neural combinatorial optimization where training often requires a differentiable loss. Third, based on the above properties, we present a simple optimization procedure which can be readily combined with existing optimization approaches to offer local improvements (i.e., the quality of the final solution is no worse than the initial solution). Finally, we also adapt our extension to optimization problems over a class of trees, such as Steiner tree and optimization-based hierarchical clustering. We present experimental results to verify our theoretical results on several combinatorial optimization problems related to permutations.

## 1 Introduction

Continuously extending combinatorial objectives is a common technique in combinatorial optimization, e.g., relaxation through linear programming, which can offer efficient optimization algorithms. Continuous extensions are particularly useful if they are differentiable, making them amenable to gradient-based optimization methods. However, it is often non-trivial to develop continuous extensions with theoretical guarantees that relate the optimization of the extension to the optimization of the combinatorial objective. In this paper, we consider combinatorial optimization problems where the goal is to minimize real-valued functions over **permutations**. We aim to develop extensions for functions on permutations with theoretical guarantees that allow for gradient-based optimization of these combinatorial objectives.

Optimization of functions on permutations is a setting that encompasses many combinatorial optimization problems with important applications. One of the most famous permutation optimization

39th Conference on Neural Information Processing Systems (NeurIPS 2025).

problems is the traveling salesperson problem (TSP), where the aim is to find the order in which a set of $n$ cities should be visited in a tour to minimize the length of the tour. TSP is an example of a vertex ordering problem, a class that contains many permutation optimization problems, in which the goal is to find an order of vertices in a graph that minimizes some objective. Examples of such problems are: directed feedback arc set (DFASP), graph cutwidth, and minimum linear arrangement (MLA). Another essential permutation optimization problem is the quadratic assignment problem (QAP), in which a bijection between $n$ facilities and $n$ locations is sought that minimizes a quadratic objective function (such bijections can be identified with permutations).

## 1.1  Our work

In this paper we develop techniques that solve combinatorial optimization problems over permutations. Formally, let $\mathcal{P}_n$ denote the set of $n \times n$ permutation matrices (w.l.o.g. permutations are viewed as matrices). For a given real-value objective function $f : \mathcal{P}_n \rightarrow \mathbb{R}$ the goal is find

$$\min f(P) \quad \text{s.t.} \quad P \in \mathcal{P}_n. \tag{1}$$

Searching over the discrete space of permutation matrices is computationally prohibitive. It therefore is appealing to turn our focus to a continuous space, such as the convex hull of permutation matrices $\mathcal{D}_n$. This space, known as the Birkhoff polytope, is the set of matrices that are doubly stochastic, i.e., have non-negative entries and rows and columns summing to one. The seminal result of Birkhoff [8] states that any matrix in the Birkhoff polytope can be decomposed into a convex combination of permutation matrices. This provides a probabilistic interpretation: any matrix $A$ in the Birkhoff polytope arises as a distribution $\nu(A)$ over permutation matrices $A = \mathbb{E}_{P \sim \nu(A)} P$. We can define our Birkhoff extension as $F(A) = \mathbb{E}_{P \sim \nu(A)}[f(P)]$ and solve

$$\min F(A) \quad \text{s.t.} \quad A \in \mathcal{D}_n. \tag{2}$$

The challenge is that a Birkhoff decomposition (the distribution $\nu(A)$) is not unique and the resulting extension may not be continuous and differentiable. Indeed, previous techniques for performing Birkhoff decomposition [8, 19] are not continuous.

**A differentiable Birkhoff extension.** Our first theoretical contribution is a Birkhoff decomposition that is continuous and almost everywhere (a.e.) differentiable (Thm. 2.4), which gives rise to an extension that is also continuous and a.e. differentiable (Property 1). Continuity is achieved by utilizing an arbitrary but fixed total order of permutations and decomposing according to this order. However, computing the extension with respect to an arbitrary ordering of permutations is intractable. To solve this, we introduce a score matrix $S$, order permutations by their inner product with $S$, and show that the resulting continuous and differentiable *Birkhoff decomposition / extension* can be efficiently computed (Property 2).

**Optimization-enabling properties.** Another appealing strength of Birkhoff extension is that minima of the extension $F$ directly correspond to minima of the function $f$ over permutations (Property 3). Moreover, Birkhoff extension admits a scheme (Property 4) for rounding a doubly stochastic matrix $A$ to a permutation $P$ that is guaranteed not to degrade the quality of the solution, i.e., $f(P) \leq F(A)$. These properties ensure that optimizing (or approximating) the extension $F$ yields optimal (or approximate) solutions to $f$.

Different choices for $S$ yield different continuous Birkhoff extensions, which is a valuable flexibility. Interestingly, we can choose any permutation $P$ (e.g, an approximate solution produced by a comparatively fast algorithm) to produce a *score matrix* (Property 4 ), which can then be combined with optimization and rounding to produce solutions that are at least as good as $P$. That is, given any existing solution to a combinatorial optimization problem, we can use it as the score matrix for a Birkhoff extension to further improve it, thereby yielding a local improvement procedure.

**Optimization algorithm.** Given the a.e. differentiability of Birkhoff extension, we can compute its gradient. However, gradient descent cannot be directly applied to optimization over the Birkhoff polytope, since after each step the resulting matrix may not be doubly stochastic. We propose a Frank-Wolfe algorithm that overcomes this issue and preserves double stochasticity throughout optimization.

Birkhoff extension is not necessarily convex and, thus, gradient-based optimization could converge to local minima. We alleviate this issue by changing the score matrix whenever the optimization

converges to a local minimum. Changing the score matrix changes the extension being optimized, potentially changing the extension to one where the current iterate is not at a local minimum, allowing further optimization. We further show that for specific changes to the score matrix (based on Property 4), we are guaranteed the quality of the rounded solution *does not decrease*.

**Application to neural combinatorial optimization.** Birkhoff extension can also be used for unsupervised neural combinatorial optimization where training often requires that the objective function is differentiable. Neural approaches are a promising new paradigm in combinatorial optimization as, unlike traditional techniques, they inherently leverage the distribution of problem instances being solved [7]. However, supervised neural combinatorial optimization is often prohibitively expensive as it requires computing exact solutions to create labels for training. Unsupervised learning, such as the set extension proposed in [29, 28], circumvents this issue by removing the need for labels. In App. B, similar to [29], we propose an unsupervised neural approach based on our Birkhoff extension. The properties of Birkhoff extensions offer advantages for unsupervised learning in that rounding guarantees ensure the minima sought in training correspond well with the combinatorial objective.

**Beyond permutation functions.** It is compelling to consider when similar techniques can be applied to other combinatorial functions. In App. C, we present analysis for applying Birkhoff extension to functions on rooted binary trees over a fixed set of leaves. Optimization of these tree functions arises in many combinatorial optimization problems such as Steiner tree [12] problems and hierarchical clustering [17].

**Experiments.** In Section 3, we perform experiments on quadratic assignment (QAP), traveling salesperson (TSP), and directed feedback arc set (DFASP) problems, showing Birkhoff extension is an effective approach for optimizing permutation functions. Our method is especially effective for the quadratic assignment problem (QAP), where it outperforms a range of Gurobi implementations and common heuristics.

To summarize, our **main contributions** are:

- A novel Birkhoff decomposition, which has continuous, a.e. differentiable, and efficiently computable coefficients.
- A continuous a.e. differentiable extension of permutation functions to real-valued functions on the Birkhoff polytope, with properties, such as rounding guarantees, desirable for optimization.
- A theoretically-justified permutation-function optimization procedure that combines this extension with gradient-based optimization.
- Combination of Birkhoff extension with an NN to yield an unsupervised neural optimizer (App. B).

## 1.2 Related work

**Extensions and optimization.** The optimization literature often focuses on building extensions with desirable optimization properties, particularly convexity and concavity [15, 38, 48]. A classical approach to extending a discrete set function $f : \{0, 1\}^n \to \mathbb{R}$ is by computing the convex closure, which is the point-wise supremum over linear functions that lower bound $f$ [22, 48]. A detailed comparison with this approach and ours is made in App. H. A prominent example of successful application of these methods to combinatorial optimization is submodular functions. The convex closure for a submodular function is identical to the Lovász extension [34], also known as the Choquet integral in decision theory [13], which leads to polynomial-time algorithms for submodular minimization [25]. A series of works [11, 53], introduced and studied multilinear extension of submodular functions which results in approximation algorithms for certain constrained submodular maximization problems. See [5] for further details on extensions of submodular functions. Convex extensions have been applied to a broader class of set functions beyond submodular ones [21], as well as to combinatorial penalties with structured sparsity [20, 40].

**Extensions and neural combinatorial optimization.** Unsupervised learning for combinatorial optimization problems has recently attracted great attention [2, 28, 46, 50]. Many frameworks based on RL [31, 33, 18, 6, 55] or supervised learning [32, 51, 24] do not hold any special requirements on the formulation of combinatorial problems. However, these approaches often suffer from dependence on labeled data or unstable training, respectively. In contrast, unsupervised learning for combinatorial optimization problems, where continuous relaxations of discrete objectives are utilized, is superior in its faster training, good generalization, and strong capability of dealing with large-scale problems.

The general idea is to use, as a loss function, a function on a continuous domain that extends the discrete function. Notable examples of these types of work are [28, 54, 9] where a probabilistic relaxation of discrete functions are used for the loss.

**Neural set function extension.** Karalias et al. [29] propose several continuous and a.e. differentiable extensions for training neural networks to optimize *set functions*, such as for the Max Clique and Max Independent Set problems. These extensions are formed by convex combinations, which allows for the efficient rounding schemes. However, many combinatorial optimization problems, such as the optimization of permutation functions, have no natural formulation as set function optimization. We use this convex decomposition framework as inspiration for our extension of permutation functions.

## 2 Birkhoff Extension

In this section we introduce our continuous and a.e. differentiable Birkhoff decomposition. We then use this decomposition to construct an extension of permutation functions to the Birkhoff polytope. Finally, we show this extension has many advantageous properties. Proofs of the claims made in this section are given in App. A.

### 2.1 Preliminaries

A *doubly stochastic $n \times n$ matrix* is one with non-negative entries where each row and column sums to 1. A *permutation matrix* is a special doubly stochastic matrix with binary entries and a single 1 in every row and column. The class of $n \times n$ doubly stochastic matrices is a convex polytope known as the Birkhoff polytope $\mathcal{D}_n$. The Birkhoff polytope lies in an $(n-1)^2$-dimensional affine subspace of $n^2$-dimensional Euclidean space defined by $2n - 1$ independent linear constraints specifying that the row and column sums all equal 1. Let $\mathcal{P}_n$ denote the set of $n \times n$ permutation matrices.

**Theorem 2.1** (Birkhoff decomposition [8]). *Any doubly stochastic matrix $A \in \mathcal{D}_n$, can be decomposed as $A = \sum_{k=1}^{M} \alpha_k P_k$ where $M < n^2 - n + 1$, $\alpha_k > 0$, $\sum_k \alpha_k = 1$, and $P_k \in \mathcal{P}_n$.*

To construct this decomposition we view $A$ as the biadjacency matrix of a bipartite graph with vertices $[n] \sqcup [n]$ and edges $(i, j)$ of weight $A(i, j)$. In this graph consider the following set of permutations, i.e. perfect matching in this graph, that do not have edges of weight zero.

**Definition 2.2.** *A permutation matrix $P \in \mathcal{P}_n$ is a perfect matching of non-negative matrix $A$ iff $P(i, j) = 1$ implies $A(i, j) > 0$. We denote the space of permutations that are perfect matchings of $A$ as $\mathcal{P}(A)$.*

The standard algorithm [8] for constructing such a decomposition is given in Alg. 1. This is an iterative algorithm that starts from $B_0 = A$ and, at each iteration $k$, takes the matrix $B_k$ resulting from the previous step, which is proportional to a doubly stochastic matrix, and finds a permutation $P$ that is a perfect matching of $B_k$. The existence of such a matching is a consequence of $B_k$ being proportional to a doubly stochastic matrix and Hall's marriage theorem. Furthermore, this matching $P$ can be computed using a standard bipartite matching algorithm in $O(n^3)$ time. If $P$ is a perfect matching of $B_k$ and $\alpha$ is the value of the smallest entry $B_k(i, j)$ in $B_k$ such that $P(i, j) = 1$, then $B_{k+1} = B_k - \alpha P$ is a matrix proportional to a doubly stochastic matrix with one less non-zero entry than in $B_k$. Note that since $P$ is a matching of $B_k$, we have $\alpha > 0$. This process is repeated until the resultant matrix is the zero matrix.

### 2.2 A continuous and a.e. differentiable Birkhoff decomposition

We extend a function $f : \mathcal{P}_n \to \mathbb{R}$ on permutations to a function $F : \mathcal{D}_n \to \mathbb{R}$ on Birkhoff polytope via the Birkhoff decomposition: $F(A) = \sum_k \alpha_k f(P_k)$. However, Birkhoff decomposition is non-unique; there may be many different ways to represent a doubly stochastic matrix as a convex combination of permutations. This non-uniqueness is evident at each step of the decomposition in the multiple choices of which permutation matrix $P \in \mathcal{P}(B_k)$ to subtract. We now describe how to fix a particular decomposition so that the coefficients $\alpha_k$, and the extension $F$, are *continuous* functions of the matrix $A$ being decomposed (with the standard $L_2$-induced topology on $\mathcal{D}_n$).

The key insight for this construction is to fix an arbitrary total order over permutation matrices and, at each step in the decomposition, always pick the valid permutation that comes first in the order.

| **Algorithm 1** Classical Birkhoff decomposition [8] | **Algorithm 2** Continuous Birkhoff decomposition |
|---|---|
| **Require:** $A \in \mathcal{D}_n$ | **Require:** $A \in \mathcal{D}_n$, identifying score matrix $S$ |
| **Ensure:** $\{(\alpha_k, P_k)\}_{k=1}^M$ s.t. $A = \sum_{k=1}^M \alpha_k P_k$, $\sum_k \alpha_k = 1$, and $\alpha_k > 0$. | **Ensure:** $\{(\alpha_k, P_k)\}_{k=1}^M$ s.t. $A = \sum_{k=1}^M \alpha_k P_k$, $\sum_k \alpha_k = 1$, and $\alpha_k > 0$. |
|    $k \leftarrow 1, B_0 \leftarrow A$ |    $k \leftarrow 1, B_0 \leftarrow A$ |
|    **while** $B_k \neq 0$ **do** |    **while** $B_k \neq 0$ **do** |
|       $P_k \leftarrow P \in \mathcal{P}(B_k)$ |       $P_k \leftarrow \text{argmax}_{P \in \mathcal{P}(B_k)} \langle P, S \rangle$ |
|       $\alpha_k \leftarrow \min_{ij}\{B_k(i,j) \mid P_k(i,j) = 1\}$ |       $\alpha_k \leftarrow \min_{ij}\{B_k(i,j) \mid P_k(i,j) = 1\}$ |
|       $B_{k+1} \leftarrow B_k - \alpha_k P_k$ |       $B_{k+1} \leftarrow B_k - \alpha_k P_k$ |
|       $k++$ |       $k++$ |
|    **end while** |    **end while** |
|    $M \leftarrow k$ |    $M \leftarrow k$ |
|    **return** $\{(\alpha_k, P_k)\}_{k=1}^M$ |    **return** $\{(\alpha_k, P_k)\}_{k=1}^M$ |

Previous decomposition algorithms fail to achieve continuity because small changes to the matrix $A$ being decomposed could change which permutation is subtracted at each step, which alter the trajectory of the decomposition. By fixing the order in which permutations are subtracted in the decomposition, we circumvent this issue. Below we introduce a continuous Birkhoff decomposition scheme, and prove its correctness (i.e., validity and continuity) in Thm. 2.4.

**Definition 2.3** (Continuous Birkhoff Decomposition). *Given an enumeration $\{P_\ell\}_{\ell=1}^{n!}$ of $\mathcal{P}_n$ (i.e, fix a total order of all permutations), and given $A \in \mathcal{D}_n$, the continuous Birkhoff decomposition of $A$ induced by $\{P_\ell\}_{\ell=1}^{n!}$ is $(\alpha_\ell, P_\ell)_{\ell=1}^{n!}$ where the coefficients are defined recurrently from $\ell = 1$ to $n!$ in order by*

$$\alpha_\ell = \min_{ij}\left\{ A(i,j) - \sum_{m=1}^{\ell-1} \alpha_m P_m(i,j) \mid P_\ell(i,j) = 1 \right\} \tag{3}$$

**Theorem 2.4.** *Given an enumeration $\{P_\ell\}_{\ell=1}^{n!}$ and given $A \in \mathcal{D}_n$, the coefficients of the continuous Birkhoff decomposition $(\alpha_\ell, P_\ell)_{\ell=1}^{n!}$ of $A$ are (i) Lipschitz continuous functions from $\mathcal{D}_n$ to $\mathbb{R}$, (ii) all non-negative and sum to 1, and (iii) yield a valid decomposition of $A$ via $A = \sum_{\ell=1}^{n!} \alpha_\ell P_\ell$. Furthermore, (iv) there are **at most** $n^2 - n + 1$ coefficients being non-zero.*

The a.e. differentiability of the continuous Birkhoff decomposition follows from Lipschitz continuity and an application of Rademacher's theorem [43].

**Theorem 2.5.** *The coefficients $\{\alpha_\ell\}_{\ell=1}^{n!}$ of the continuous Birkhoff decomposition are almost everywhere differentiable functions from $\mathcal{D}_n$ to $\mathbb{R}$.*

Now that we have constructed a continuous Birkhoff decomposition, the next questions are how to represent that total order of all permutation matrices, and how to compute this decomposition efficiently. Indeed, for an arbitrary ordering of permutations, efficient computation is not feasible as it requires referencing the order of $n!$ elements. We overcome this challenge by instead focusing on orderings of permutations that arise from an inner product.

**Definition 2.6** (Score-Induced Birkhoff Decompositions). *Given an $n \times n$ matrix $S$, the score of a permutation $P$ is $\langle S, P \rangle = \sum_{ij} S(i,j)P(i,j)$. We call $S$ a score matrix, and say that $S$ is identifying if it assigns a unique score to every permutation, thereby inducing a total order on $\mathcal{P}_n$.*

*Furthermore, given an identifying score matrix $S$, the Birkhoff decomposition as specified in Def. 2.3 with respect to an ordering of permutations by their score $\{P_\ell\}_{\ell=1}^{n!}$ is called an $S$-induced Birkhoff decomposition.*

A simple example of an identifying score matrix is given by $S(i,j) = 2^{(i+nj)}$. The score-matrix induced total order is particularly effective because for any $A \in \mathcal{D}_n$, the permutation $P \in \mathcal{P}(A)$ that comes first in this order can be found efficiently by solving a maximum weight matching problem. Consequently, decompositions with respect to this order can be constructed efficiently:

**Theorem 2.7.** *Given an identifying score matrix $S$, the $S$-induced Birkhoff decomposition can be computed in $O(n^5)$ time by Alg. 2.*

In practice, we may wish to use score matrices other than $S(i,j) = 2^{(i+nj)}$. The following theorem shows that random assignment of $S$ is sufficient for $S$ to be identifying.

**Claim 2.8.** *If the entries $S$ are independent absolutely continuous random variables $S(i,j) \in \mathbb{R}$ then $S$ is identifying almost surely.*

## 2.3 Properties of Birkhoff extension

We present several properties that make our score-induced Birkhoff extension a desirable candidate for optimization. Proofs of these properties are given in App. A.

**Definition 2.9.** *Given $A \in \mathcal{D}_n$ and an ordering of permutations $\{P_\ell\}_{\ell=1}^{n!}$, let $(\alpha_k, P_k)_{k=1}^M$ be the non-zero Birkhoff coefficients defined in Def. 2.3. For any $f : \mathcal{P}_n \to \mathbb{R}$, the Birkhoff extension of $f$ is the function $F : \mathcal{D}_n \to \mathbb{R}$ where*

$$F(A) = \sum_{k=1}^M \alpha_k f(P_k). \tag{4}$$

*We say $F$ is* score induced *or $S$-induced if the ordering of permutations is induced by $S$. We sometimes emphasize the dependence on $S$ by using $F_S$ to denote the $S$-induced Birkhoff extension.*

Almost everywhere differentiability and continuity of $F$ are essential for gradient-based optimization. These properties follow from continuity and a.e. differentiability of the coefficients $\{\alpha_\ell\}_{\ell=1}^{n!}$. Furthermore, we show that when computing the gradient of $F$ one only needs to consider the non-zero terms in the Birkhoff decomposition.

**Property 1.** *Birkhoff extensions are Lipschitz continuous and almost everywhere differentiable. Furthermore if $L_+ = \{\ell \in [n!] : \alpha_\ell > 0\}$, then $\nabla_A F(A) = \sum_{\ell \in L_+} (\nabla_A \alpha_\ell) f(P_\ell)$ almost everywhere.*

Computing a Birkhoff extension reduces to computing the corresponding Birkhoff decomposition, thus, Alg. 2 gives efficient computation of score-induced Birkhoff extensions (see Thm. 2.7).

**Property 2.** *Score-induced Birkhoff extensions $F$ can be computed in $O(n^5)$ time.*

One concern when optimizing a continuous extension to a combinatorial function $f$ is that the optimization reaches some minimum in the extended space that does not correspond with minima of the combinatorial function (which is our true goal). Property 3 below shows that (global) minima of the extension $F$ (i.e., over $\mathcal{D}_n$) are related to those of $f$ over the permutations.

**Property 3.** *Let $F$ be a Birkhoff extension of $f : \mathcal{P}_n \to \mathbb{R}$. Then (1) $\min_{P \in \mathcal{P}_n} f(P) = \min_{A \in \mathcal{D}_n} F(A)$ and (2) $\operatorname{argmin}_{A \in \mathcal{D}_n} F(A) \subseteq \operatorname{Conv}(\operatorname{argmin}_{P \in \mathcal{P}_n} f(P))$.*

The Birkhoff decomposition leads to a simple rounding strategy:

**Definition 2.10.** *Given a matrix $A \in \mathcal{D}_n$ and a score matrix $S$ that induces a Birkhoff decomposition $(\alpha_k, P_k)_{k=1}^M$, we define*

$$\operatorname{round}_S(A) = \operatorname{argmin}_{P_k : k \in [M]} (f(P_k)). \tag{5}$$

Note that $\operatorname{round}_S(A)$ can be computed in $O(n^5)$ time by computing a Birkhoff decomposition of $A$. The rounding scheme is lossless in that it can *only improve* solution quality; see Property 4-1 below. Consequently, optimizing $f$ reduces to optimizing the Birkhoff extension $F$, as any minimum of $F$ can be used to derive a minimum of $f$. Furthermore, approximations to $\min_{A \in \mathcal{D}_n} F_S(A)$ can be rounded to approximations for $\min_{P \in \mathcal{P}_n} f(P)$.

**Property 4.** *Let $F_S$ be a score-induced Birkhoff extension of $f : \mathcal{P}_n \to \mathbb{R}$. For any $A \in \mathcal{D}_n$, then*

1. *$f(\operatorname{round}_S(A)) \le F_S(A)$. Furthermore, if $A$ is a $C$-approximation for $\min_{A \in \mathcal{D}_n} F_S(A)$, then $\operatorname{round}_S(A)$ is a $C$-approximation for $\min_{P \in \mathcal{P}_n} f(P)$.*
2. *If $P^* \in \mathcal{P}_n$ with $\max_{ij} |P^*(i,j) - S(i,j)| < \frac{1}{2n}$, then $f(\operatorname{round}_S(A)) \le f(P^*)$.*

Another useful quality of this rounding scheme, Property 4-2, is that if $S$ is sufficiently close to some permutation $P$, then rounding always yields a solution at least as good as $P$. This holds *independent of the matrix $A$ that is being rounded* and gives useful flexibility to Birkhoff extension optimization. In particular if the score $S$ is close to an approximate solution to the combinatorial optimization problem, then rounding always produces a solution at least as good as the approximation. For example, if $S = P_{\text{approx}} + Q$ where $P_{\text{approx}}$ is solution produced by a fast approximation algorithm and $Q$ is a

**Algorithm 3** Static score Frank-Wolfe over $\mathcal{D}_n$

**Require:** $f : \mathcal{P}_n \to \mathbb{R}$, random $A \in \mathcal{D}_n$, score $S$

    **for** $t = 1 \cdots T$ **do**
        $P_t \leftarrow \operatorname{argmax}_{P \in \mathcal{P}_n} \langle \nabla F_S(A_t), P \rangle$
        $A_{t+1} \leftarrow (1 - \lambda_t) A_t + \lambda_t P_t$
    **end for**
    **return** $\operatorname{round}_S(A_T)$

**Algorithm 4** Dynamic score Frank-Wolfe over $\mathcal{D}_n$

**Require:** $f : \mathcal{P}_n \to \mathbb{R}$, random $A \in \mathcal{D}_n$, score $S$

    **for** $t = 1 \cdots T$ **do**
        $P_t \leftarrow \operatorname{argmax}_{P \in \mathcal{P}_n} \langle \nabla F_S(A_t), P \rangle$
        $A_{t+1} \leftarrow (1 - \lambda_t) A_t + \lambda_t P_t$
        $\mathcal{P}^* \leftarrow \mathcal{P}^* \cup \operatorname{Birkhoff}(A_{t+1})$
        **if** update_score **then**
            $Q \sim \operatorname{Unif}([0, 1]^{n \times n})$
            $P^* \leftarrow \frac{1}{2n} Q + \operatorname{argmin}_{P \in \mathcal{P}^*} f(P)$
            $S \leftarrow P^*$
        **end if**
    **end for**
    **return** $\operatorname{round}_S(A_T)$

random noise matrix with entries in $[0, 1/n^2]$. (Adding $Q$ ensures that $S$ is almost surely identifying by Claim 2.8.) Then, gradient-based optimization of the Birkhoff extension associated with $S$ finds solutions that are guaranteed to be no worse than the approximation $P_{\text{approx}}$. This means that we can use Birkhoff extension as a *local improvement strategy* to potentially improve any given solution. It is important that $P_{\text{approx}}$ is used as the **score matrix**, not as an initialization for the optimized matrix $A \in \mathcal{D}_n$. In fact, initializing $A$ at $P_{\text{approx}}$ yields weaker guarantees as optimization may produce a solution of worse quality in this case.

## 2.4 Optimization procedure with dynamic score

Consider the optimization problem $\min_{P \in \mathcal{P}_n} f(P)$ where the goal is to minimize a function $f : \mathcal{P}_n \to \mathbb{R}$ on permutations. A natural relaxation for this optimization problem is to optimize the Birkhoff extension of $f$ over the set of doubly stochastic matrices; namely, the constrained optimization problem of the form $\min_{A \in \mathcal{D}_n} F(A)$. Here we propose an iterative first-order optimization algorithm that is concerned with optimizing an objective function over the Birkhoff polytope $\mathcal{D}_n$.

**Frank-Wolfe.** One difficulty in gradient-based approaches to optimize constrained optimization problems is the risk of stepping outside of the feasible region. We address this difficulty by adapting the famous Frank-Wolfe approach [23]. In contrast to *projected gradient descent* approaches, the idea is to follow a direction of descent that is best aligned with the negative of the gradient for which we can also easily ensure feasibility. This is done via optimizing the negative of the gradient over the extreme vertices $\mathcal{P}_n \subset \mathcal{D}_n$ and then taking the obtained permutation as an alternative direction of descent. The overall process is outlined in Alg. 3. Moreover, Frank–Wolfe reduces to iterated linear minimizations for which highly efficient combinatorial algorithms exist, whereas projection-based methods require solving quadratic programs over the Birkhoff polytope. In particular, $\operatorname{argmin}_{P \in \mathcal{P}_n} \langle \nabla F_S(A_t), P \rangle$ can be computed by finding a maximum weight matching in the bipartite graph $G$ that has vertices $[n] \sqcup [n]$ and an edge from vertex $i$ to vertex $j$ of weight $\nabla F_S(A_t)(i, j)$; the weight of a matching $P$ in $G$ is $\langle \nabla F_S(A_t), P \rangle$. Similar applications of Frank-Wolfe to continuous optimization over the Birkhoff polytope have been discussed by Tewari et al. [49] and Jaggi [27], however these works do not consider applications to combinatorial optimization, as we do.

**Dynamic score.** One issue with gradient-based optimization of a Birkhoff extension is that the extension is not necessarily convex, and thus, an optimization algorithm may converge to local minima. Even if the function is convex, analyzing the convergence of Frank-Wolfe type algorithms for non-smooth functions is highly nontrivial [3, 42, 4] and, in general, they may not converge at all, see [39] for a counterexample. To alleviate these issues, we propose an optimization scheme in which the score matrix $S$ is frequently changed to help escape local minima. The key to the efficacy of this approach is that the score matrix can be changed without decreasing the quality of the rounded solution. Property 4-2 gives conditions for this change to be made without harming the optimization. In particular, this occurs under the condition that the new score matrix $S'$ is sufficiently close to a permutation $P$ satisfying $f(P) \leq f(\operatorname{round}_S(A))$. We further theoretically validate this approach by showing at any $A \in \mathcal{D}_n$ there is a score matrix $S$ such that $A$ is not a local minimum of $F_S$ (App. G).

**Theorem 2.11** (Escaping Local Minima)**.** *Let $f : \mathcal{P}_n \to \mathbb{R}$ be any function on permutations, there exists a score matrix $S$ such that $A$ is not a* local *minimum of $F_S$.*

As finding such a score matrix is computationally hard, we use the following procedure in practice. Let $\text{Birkhoff}_S(A)$ be the permutations with non-zero coefficients in the $S$-induced Birkhoff decomposition of $A$ and for each iterate $A_t$ let $\mathcal{P}^* = \cup_{0 \le t' \le t} \text{Birkhoff}_S(A_{t'})$. We update the score to $S' = \arg\min_{P \in \mathcal{P}^*} f(P) + \frac{1}{2n}Q$ where $Q$ is a matrix with uniform random entries in $[0,1]$. By Property 4-2 this update satisfies $f(\text{round}_{S'}(A_t)) \le f(\text{round}_S(A_t))$. This procedure is outlined in Alg. 4, where `update_score` is a flag determined externally indicating if convergence has occurred and the score should be updated.

**Adaptations to constraints and trees.** We provide further extensions based on our Birkhoff decomposition. First, we show how Birkhoff extension can be adapted to a class of rooted binary trees over a fixed leaf set (App. C). Next, we show that Birkhoff extension can easily handle a broad class of simple constraint (App. I). In particular, we consider constraints that can be incorporated into the corresponding matching problem.

**Unsupervised learning.** We apply Birkhoff extension to train neural combinatorial optimization solvers (App. B). As Birkhoff extension is a.e. differentiable, it can be used as a loss function in unsupervised learning. Using this loss, neural networks can be trained to output solutions to combinatorial problems. Our proof-of-concept experiments show that training a neural network using Birkhoff extension yields a model that predicts good approximate solutions. In particular, the output of this NN can be used to initialize our Birkhoff extension optimization algorithm to provide substantial speedups. Furthermore, this generalizes to larger problem instances than seen in training.

## 3  Experiments

We carry out experiments on three different combinatorial optimization problems: the quadratic assignment problem (QAP), the (Euclidean) traveling salesman problem (TSP), and the directed feedback arc set problem (DFASP). For QAP, we seek a bipartite matching that minimizes a quadratic loss. For TSP, the goal is to find the shortest possible tour visiting each city once and returning to the starting point. In DFASP, we aim to find a vertex ordering that minimizes the number of edges directed against the order. Detailed definitions and linear integer programming formulations for each problem are given in App. D. For each problem instance, we apply a variant of Alg. 4 to optimize and compare its performances with baselines. In particular, we update the score matrix every 10 epochs (i.e., `update_score` = True in Alg. 4 if and only if $t \in \{10, 20, \ldots, T\}$). To overcome the $O(n^5)$ time complexity, we truncate the Birkhoff decomposition to only the first $k = 5$ terms. Automatic differentiation is used to compute gradients. Additional experiments and ablations are given in App. F.

**Summary of results.** We first summarize our results and then provide specific discussion (full details in App. E). Our optimization algorithm performs best at QAP, where it outperforms two different Gurobi implementations and two popular heuristic approaches (which are the default `scipy` heuristics). We suspect Birkhoff extension performs particularly well here since permutation matrices naturally model matchings in that the $i, j$ matrix entry indicates a match between facility $i$ and location $j$. Other permutation problems are less amenable to this representation; e.g, in TSP the $i, j$ matrix entry indicates that city $i$ occurs $j$th in the tour. Indeed, Gurobi is highly effective for TSP and outperforms our Birkhoff extension-based approach both in terms of accuracy and efficiency. Nevertheless, we show that Birkhoff extension can still be used as a local improvement to further improve solutions provided by the minimum spanning tree (MST) approximation algorithm or by quadratic programming. Gurobi does not perform well for DFASP. In particular, our approach provides superior solutions for large instances for DFASP.

**QAP.** We evaluate on the QAPLIB library [10], which contains $N = 136$ problem instances with sizes between 12 and 256. Some instances have confirmed optimal objective values, while others have the best-known objective values reported. For Gurobi, we use the Kaufman–Broeckx linearization [30] for the Mixed-Integer Linear Programming (MILP), and we set same maximum time budget for Gurobi and our method as $2n$ seconds, and we report the actual run-time on average for all methods. For the same time budget, the canonical ILP formulation [56] fails on 70% of instances. We also test the same benchmark with two other heuristics: Fast Approximate QAP [52] and 2-opt algorithm [16]. Our method outperforms all benchmarks (see results in Table 1).

**TSP.** We generate instances by uniformly sampling $n$ vertices $v_i \in [0,1]^2$. For each size $n \in \{20, 30, 40, 50, 100\}$, we generate $N = 50$ instances. Comparisons are made with an MST-based

| Method | QAP: Assignment Cost ↓ | |
|---|---|---|
| | Average Gap from the best known | Run-Time (secs) |
| Gurobi, Kaufman-Broeckx | 7.67% | 90.3 |
| Fast Approximate QAP [52] | 15.41% | 0.01 |
| 2-opt [16] | 10.38% | 22.82 |
| Random $S$ Init. Alg. 4 | **6.30%** | 23.01 |

Table 1: Performances of methods for QAP in terms of assignment costs. As QAPLIB contains the best known solutions for each problem instance, we report the average gap to these objective values.

approximation algorithm [14], a quadratic programming (QP) relaxation, and Gurobi (which is optimal). For Alg. 4 we test three different score matrix $S$ initializations: uniformly random in $[0, 1]^{n^2}$, MST approximation, and the QP relaxation solution. The learning rate is $\eta = 0.01$. The maximum number of optimization steps is $T = 10000$, and early stopping triggers after 2000 steps without improvement. All optimization processes converge (see optimization curves for different heuristics in App. E). Results for TSP are presented in Table 2. Although Gurobi is highly effective and returns the optimal solution, our approach can still provide local improvements over QP and MST (see the last two rows in Table 2) by using those solutions as the input score matrix for Alg. 4.

**DFASP.** We generate instances using a directed *Erdős-Rényi* model with $p \in \{0.1, 0.5, 0.9\}$. Again, for each problem size $n \in \{20, 50, 100\}$, we generate $N = 50$ instances. Note, Gurobi is not able to find an optimal solution within a reasonable time for DFASP. Hence, to compare the quality of Gurobi vs. our approach, we limit the runtime of both algorithms to $\frac{n}{10}$ minutes for equal comparison. Consequently, the number of steps $T$ varies for different problem instances. (Additional experiments (App. E) show these optimization algorithms offer minimal improvement after the $\frac{n}{10}$ minute time limit.) The learning rate is $\eta = 0.005$. Results, given in Table 3, show our method is competitive with Gurobi and shows superior performance for larger instances.

| Method | TSP: Tour Length ↓ | | | | |
|---|---|---|---|---|---|
| $n =$ | 20 | 30 | 40 | 50 | 100 |
| Gurobi | **3.889** | **4.531** | **5.190** | **5.707** | **7.748** |
| MST | 4.746 | 5.784 | 6.835 | 7.410 | 10.288 |
| QP | 4.553 | 5.666 | 6.818 | 7.782 | 11.896 |
| MCTS w. Rand. Simulation | 4.558 (0.5s) | 5.563 (1.7s) | 6.706 (4.7s) | 7.468 (9.8s) | 11.781 (113.37s) |
| Random $S$ Init. Alg. 4 | 4.518 | 5.681 | 7.139 | 8.433 | 15.615 |
| MST $S$ Init. Alg. 4 (Improv.) | 4.345 (8.33 %) | 5.290 (8.53 %) | 6.328 (7.42 %) | 6.892 (6.99 %) | 9.836 (4.60 %) |
| QP $S$ Init. Alg. 4 (Improv.) | 4.405 (3.25 %) | 5.633 (0.58 %) | 6.709 (1.60 %) | 7.670 (1.45 %) | 11.782 (0.96 %) |

Table 2: Performances of methods for TSP in terms of tour length. Best results are marked as bold; second bests are blue. If score is initialized to an approximate solution, percentages indicate proportional improvement of Alg. 4 over initialization. See appendix E for optimization

| Method | DFASP: Feedback-Arc Set Size ↓ | | | | | | | | | | | |
|---|---|---|---|---|---|---|---|---|---|---|---|---|
| | $p = 0.1$ | | | | $p = 0.5$ | | | | $p = 0.9$ | | | |
| $n =$ | 20 | 50 | 100 | 250 | 20 | 50 | 100 | 250 | 20 | 50 | 100 | 250 |
| Gurobi | **3.8** | **44.8** | **269.1** | **875.6** | **67.7** | **475.4** | 2229.9 | 6831.7 | **156.7** | **1028.0** | 4362.7 | 13652.4 |
| Random $S$ Init. Alg. 4 | 4.7 | 53.0 | 289.0 | 991.3 | 65.0 | 496.5 | **2123.4** | **6459.8** | 157.5 | 1039.0 | **4253.7** | **12198.7** |

Table 3: Performances of DFASP algorithms in terms of feedback-arc set cardinality.

# 4 Concluding remarks

We present Birkhoff extension, a continuous a.e. differentiable extension of permutation functions to doubly stochastic matrices, which has rounding guarantees. Combining this extension with a gradient-based optimization algorithm, we develop an iterative optimization framework for permutation functions. We present experiments to validate our approach for combinatorial optimization problems.

We propose a neural optimizer based on our Birkhoff extension, however, this direction requires further exploration and far more extensive experiments. We leave this as a future direction to investigate. We also note that currently we use a simple strategy to update the score matrix. It would be interesting to explore more effective update strategies. Finally, computing Birkhoff extension is expensive ($O(n^5)$ time complexity), although in practice, the number of permutations is usually far fewer than $n^2$. It will be interesting to investigate how to improve the time complexity, or how to obtain an updated the Birkhoff decomposition efficiently as the input matrix changes, given that one needs to compute this decomposition many times within our optimization framework. One potential for speedup is to develop matching algorithms that are more amenable to GPU computation. We initiate discussion by introducing a GPU friendly matching algorithm in App. L.

## Acknowledgments

This work is partially supported by NSF (National Science Foundation) by grant CCF-2112665.

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

## Table of Contents

## Contents

## A   Deferred Proofs

*proof of Thm. 2.4.* We show continuity (i) by induction. Note that Eq. 3 makes reference to the order $\{P_\ell\}_{\ell=1}^{n!}$, which we now induct on. For the base case, we have that $\alpha_1$ is a Lipschitz continuous function of $A$ as the sum in Eq. 3 disappears. Now, assume $\alpha_m$ for $m < \ell$ is a Lipschitz continuous function of $A$. Then, by Eq. 3, $\alpha_\ell$ is a Lipschitz continuous function of $A$, as $\min$ is a Lipschitz continuous function.

Now for each $\ell \in [n!]$, set $B^\ell = A - \sum_{m=1}^{\ell} \alpha_m P_m$. Clearly, we have $B^\ell = B^{\ell-1} - \alpha_\ell P_\ell$, and $\alpha_\ell = \min_{i,j}\{B^{\ell-1}(i,j) \mid P_\ell(i,j) = 1\}$. Furthermore let $Z^\ell$ denote the set of indices $(i,j)$ of non-zero entries in $B^\ell$; that is, $Z^\ell = \{(i,j) \mid B^\ell(i,j) > 0\}$. First, note that since $A$ and every $P_m$ are doubly stochastic, $B^\ell$ must be proportional to some doubly stochastic matrix for any $\ell$. Indeed, for any $\ell$ the matrix $B^\ell$ has rows and columns that sum to $1 - \sum_{m=1}^{\ell} \alpha_m$. Next, we use induction to show that (a) $B^\ell$ has non-negative entries, (b) $\alpha_\ell = 0$ or $\alpha_\ell > 0$, and (c) if $\alpha_\ell > 0$, then $Z^\ell$ is a strict subset of $Z^{\ell-1}$; i.e. $Z^\ell \subset Z^{\ell-1}$. Note that (c) implies that the number of non-zero entries in $B^\ell$ is strictly smaller than that in $B^{\ell-1}$ whenever $\alpha_\ell > 0$. We then use these properties to prove statements (ii) – (iv) in the theorem.

First, it is easy to see that properties (a) - (c) hold for the base case $\ell = 1$. Now consider $\ell > 1$, and assume they hold for any $r < \ell$. Since all entries in $B^{\ell-1} = A - \sum_{m=1}^{\ell-1} \alpha_m P_m$ are non-negative, namely $B^{\ell-1}(i,j) = A(i,j) - \sum_{m=1}^{\ell-1} \alpha_m P_m(i,j) \geq 0$ for any $i, j \in [n]$, we have $\alpha_\ell \geq 0$, proving property (b).

If $\alpha_\ell = 0$, then properties (a) and (c) trivially hold so assume $\alpha_\ell > 0$. Let $i^*, j^*$ be indices that give rise to $\alpha_\ell$. That is, $\alpha_\ell = B^{\ell-1}(i^*, j^*)$ and $P_\ell(i^*, j^*) = 1$. By definition of $\alpha_\ell$, for any other $i, j$ such that $P_\ell(i,j) = 1$, we have that $\alpha_\ell \leq B^{\ell-1}(i,j)$. Since $P_\ell$ is a binary matrix, this means that for all $i, j \in [n]$, $\alpha_\ell P_\ell(i,j) \leq B^{\ell-1}(i,j)$. Hence all entries in $B^\ell = B^{\ell-1} - \alpha_\ell P_\ell$ are non-negative, proving property (a).

Furthermore, note that by construction, $B^\ell(i^*, j^*) = B^{\ell-1}(i^*, j^*) - \alpha_\ell P_\ell(i^*, j^*) = 0$, while $B^{\ell-1}(i^*, j^*) = \alpha_\ell > 0$. As each entry in $B^\ell$ is less than the corresponding entry in $B^{\ell-1}$, but are still non-negative, we can conclude $Z^\ell \subset Z^{\ell-1}$. In particular, $(i^*, j^*) \in Z^{\ell-1}$ but not in $Z^\ell$. This proves property (c).

Hence by induction, properties (a), (b) and (c) hold for all $\ell \in [n!]$. We are now ready to prove statements (ii) - (iv).

To prove (iii) we aim to show that $B^{n!} = 0$. We proceed by contradiction so suppose $B^{n!} \neq 0$. Then $B^{n!}$ is proportional to a doubly stochastic matrix and must have at least one matching, say $P_{\ell^*}$, by Hall's Marriage theorem. However, as all entries in $B^\ell$ are non-decreasing as $\ell$ increases, the set of non-zero entries in $B^{\ell^*-1}$ is a super set of those in $B^{n!}$ and, thus, super set of those in $P_{\ell^*}$. Therefore we must have $\alpha_{\ell^*} > 0$. Furthermore, let $(i^*, j^*)$ be the pair of indices giving rise to $\alpha_{\ell^*}$, i.e., $\alpha_{\ell^*} = B^{\ell^*-1}(i^*, j^*)$ and $P_\ell(i^*, j^*) = 1$. Then $B^{\ell^*}(i^*, j^*) = B^{\ell^*-1}(i^*, j^*) - \alpha_{\ell^*} P_{\ell^*}(i^*, j^*) = 0$. As all entries can only decrease, $B^{n!}(i^*, j^*) = 0$. However, since $P_{\ell^*}(i^*, j^*) = 1$, this means that $P_{\ell^*}$ cannot be a matching for $B^{n!}$, which is a contradiction. Hence our assumption that $B^{n!}$ is non-zero cannot be true, and we must have that $B^{n!} = 0$, meaning $A = \sum_{\ell=1}^{n!} \alpha_\ell P_\ell$. This proves statement (iii).

To prove (ii), note that by (b), all $\alpha_\ell \geq 0$. Furthermore, as we mentioned earlier, for any $\ell$ the matrix $B^\ell$ has rows and columns summing to $1 - \sum_{m=1}^{\ell-1} \alpha_m$. Since $B^{n!}$ is the zero-matrix, it follows that $1 - \sum_{m=1}^{n!} \alpha_m = 0$, proving (ii).

Finally, we bound the number of non-zero coefficients used in the decomposition and prove statement (iv). Consider an $\ell^*$ for which there are $n^2 - n + 1$ non-zero coefficients $\alpha_m$ with $m \leq \ell^*$. If no such $\ell^*$ exists we are done. If such $\ell^*$ exists, then $B^{\ell^*} = A - \sum_{m=1}^{\ell^*} \alpha_m P_m$ is either the zero matrix or is proportional to some doubly stochastic matrix. If $B^{\ell^*}$ is the zero matrix, then we are done. So assume $B^{\ell^*}$ proportional to some doubly stochastic matrix. Note that any doubly stochastic matrix necessarily has at least $n$ non-zero entries (as each row-sum needs to be 1). Hence $B^{\ell^*}$ has at least $n$ non-zero entries. However, as the set of indices for non-zero entries $Z^m$ strictly decreases each time $\alpha_m > 0$ (property (c) proved above), and there are $n^2 - n + 1$ such $\alpha_m$ with $m \leq \ell^*$, this means that

the number of non-zero entries in the original matrix $A$ is at least $n^2 - n + 1 + n = n^2 + 1$, which is not possible (as there are only $n^2$ entries in a $n \times n$ matrix). Hence $B^{\ell^*}$ must be a zero matrix, and there cannot be more than $n^2 - n + 1$ non-zero coefficients in $\{\alpha_\ell\}_{\ell=1}^{n!}$. This completes the proof of statement (iv). $\qquad\square$

*proof of Thm. 2.5.* $\mathcal{D}_n$ lies in an $(n-1) \times (n-1)$ dimensional affine subspace of $\mathbb{R}^{n^2}$ defined by all $n \times n$ matrices with rows and columns summing to 1. We can parameterize this subspace with the linear map

$$\phi : \mathbb{R}^{(n-1)^2} \to \mathbb{R}^{n^2} \tag{6}$$

$$\begin{pmatrix} a_{1,1} & \cdots & a_{1,n-1} \\ \vdots & \ddots & \vdots \\ a_{n-1,1} & \cdots & a_{n-1,n-1} \end{pmatrix} \mapsto \begin{pmatrix} a_{1,1} & \cdots & a_{1,n-1} & 1 - \sum_{j=1}^{n-1} a_{1,j} \\ \vdots & \ddots & \vdots & \vdots \\ a_{n-1,1} & \cdots & a_{n-1,n-1} & \vdots \\ 1 - \sum_{i=1}^{n-1} a_{i,1} & \cdots & \cdots & 1 - \sum_{i=1}^{n-1}\left(1 - \sum_{j=1}^{n-1} a_{i,j}\right) \end{pmatrix} \tag{7}$$

Note that $X = \phi^{-1}(\mathcal{D}_n)$ is the subset of $R^{(n-1)\times(n-1)}$ containing matrices with rows and column sums in $[0, 1]$. This is a closed set whose boundary has measure zero since it has dimension less than $(n-1) \times (n-1)$. The Lipschitz continuity of $\phi$ and $\alpha_\ell$ gives that $\alpha_\ell \circ \phi$ is Lipschitz continuous. Applying Rademacher's theorem [43] to the interior $\text{int}(X)$ of $X$ yields that $\alpha_\ell \circ \phi$ is an a.e. differentiable function from $\text{int}(X) \to \mathbb{R}$. Since $X \setminus \text{int}(X)$ has measure zero, we have that $\alpha_\ell \circ \phi$ is an a.e. differentiable function on $X$. Let $N \subset X$ be the set of points for which $\alpha_\ell \circ \phi$ is not differentiable. Then $\phi(N)$ is the set of points such that $\alpha_\ell$ is not differentiable. Furthermore, $\phi(N)$ has Lebesgue measure zero in $\mathcal{D}_n$ since $N$ has measure zero and $\phi$ is a surjective linear map to $\mathcal{D}_n$. $\qquad\square$

*proof of Thm. 2.8.* Consider two distinct permutations $P, P' \in \mathcal{P}_n$. Let $I$ and $I'$ contain the pairs $(i, j)$ of indices such that $P(i, j) = 1$ and $P'(i, j) = 1$ respectively. The score of these permutations $\langle P, S \rangle = \sum_{(i,j)\in I} S(i, j)$ and $\langle P', S \rangle = \sum_{(i,j)\in I'} S(i, j)$, are equal if and only if

$$\sum_{(i,j)\in I\setminus I'} S(i, j) = \sum_{(i,j)\in I'\setminus I} S(i, j). \tag{8}$$

The left-hand side and the right-hand side of this equation are independently distributed absolutely continuous random variables, so they are equal with probability zero. Since any pair of permutations have different scores almost surely, all permutations have different scores almost surely by the union bound. $\qquad\square$

*proof of Property 1.* Lipschitz continuity and a.e. differentiability of $F$ follow from the Lipschitz continuity and a.e. differentiability of $\{\alpha_k\}_{k=1}^{M}$ (theorems 2.4 and 2.5).

Now we show that the gradient $\nabla_A F(A)$ is equivalent to

$$\nabla_A F(A) = \sum_{k=1}^{M} (\nabla_A \alpha_k) f(P_k). \tag{9}$$

We start with a dimensionality argument showing there are never more than $n^2 - 2n + 2$ terms in a Birkhoff decomposition. First we show the set $\{P_k\}_{k=1}^{M}$ of permutations with positive coefficients is linearly independent.

Suppose some $P_\ell \in \{P_1, \ldots, P_M\}$ can be written as a linear combination $P_\ell = \sum_{k\in[M]\setminus\{\ell\}} c_k P_k$ for $c_k \in \mathbb{R}$. We first show all $c_k$ with $k < \ell$ must be zero. The proof is by induction. We have $c_1 = 0$ as there must be some $i, j \in [n]$ such that $P_1(i, j) = 1$ but $P_k(i, j) = 0$ for $k > 1$. Now consider some $k < \ell$ and suppose $c_1 = \ldots = c_{k-1} = 0$. Then $c_k = 0$ as there is a $i, j \in [n]$ such that $P_k(i, j) = 1$ but for $k' > k$, $P_{k'}(i, j) = 0$. We can then conclude $c_1 = \ldots = c_{\ell-1} = 0$. Finally, since there is $i, j \in [n]$ such that $P_\ell(i, j) = 1$ but $P_k(i, j) = 0$ for $k > \ell$ we can conclude

$P_\ell = \sum_{k \in [M] \setminus \{\ell\}} c_k P_k = \sum_{k=\ell+1}^{M} c_k P_k$ cannot hold. We have, thus, shown that the set of permutations with positive coefficients is linearly independent.

Since this set of permutations is linearly independent, the maximum number of permutations with positive coefficient is one more than the dimension of $\mathcal{D}_n$ which is $n^2 - 2n + 2$. Suppose that $A$ has a full Birkhoff decomposition, that is, there are $n^2 - 2n + 2$ positive terms in its Birkhoff decomposition. Then, by continuity of $\{\alpha_\ell\}_{\ell=1}^{n!}$, there is an open ball in $\mathcal{D}_n$ containing $A$ such that all decompositions of points in the ball have the same positive coefficients. Thus, the gradient of the zero coefficients at $A$ is zero.

It now remains to show that a.e. $A \in \mathcal{D}_n$ has a full Birkhoff decomposition. For each subset $E \subset \mathcal{P}_n$ with $|E| = n^2 - 2n + 1$, note that the convex hull $\mathrm{Conv}(E)$ of $E$ has dimension $n^2 - 2n$ and, therefore, has measure zero in the space $\mathcal{D}_n$, which has dimension $n^2 - 2n + 1$. Consider the union of such convex hulls

$$\mathcal{E} = \bigcup_{\substack{E \subset \mathcal{P}_n \\ |E| = n^2 - 2n + 1}} \mathrm{Conv}(E). \tag{10}$$

The space $\mathcal{E}$ also has measure zero. Suppose $A$ does not have full Birkhoff decomposition. Then $A \in \mathcal{E}$ as it can be represented as the convex combination of at most $n^2 - 2n + 1$ permutation matrices. We can then conclude that a.e. $A \in \mathcal{D}_n$ has a full Birkhoff decomposition.

$\square$

*proof of Property 3.* (1.) Note that for $P \in \mathcal{P}_n$ we have $f(P) = F(P)$ since all Birkhoff coefficients of $P$ have only one term. Thus, $\min_{P \in \mathcal{P}_n} f(P) \geq \min_{A \in \mathcal{D}_n} F(A)$. Now suppose $\min_{P \in \mathcal{P}_n} f(P) > \min_{A \in \mathcal{D}_n} F(A)$ and let $A \in \mathrm{argmin}_{A \in \mathcal{D}_n} F(A)$. Note $F(A)$ is a convex combination $F(A) = \sum_{k=1}^{M} \alpha_k f(P_k)$ and $\sum_{k=1}^{M} \alpha_k = 1$. We can then conclude that since $\min_{P \in \mathcal{P}_n} f(P) > \sum_{k=1}^{M} \alpha_k f(P_k)$ there must be some $P_k$ such that $f(P_k) \leq \min_{P \in \mathcal{P}_n} f(P)$, a contradiction.

(2.) Suppose $A$ minimizes $F(A)$ over $\mathcal{D}_n$. Then $F(A) = \min_{P \in \mathcal{P}_n} f(P)$, which occurs only if for each $P_k$ in the convex combination $F(A) = \sum_{k=1}^{M} \alpha_k f(P_k)$ we have $f(P_k) = \min_{P \in \mathcal{P}_n} f(P)$. Since $A = \sum_{k=1}^{M} \alpha_k P_k$ and $P_k \in \mathrm{argmin}_{P \in \mathcal{P}_n} f(P)$ the claim $\mathrm{argmin}_{A \in \mathcal{D}_n} F(A) \subseteq \mathrm{Conv}(\mathrm{argmin}_{P \in \mathcal{P}_n} f(P))$ holds. $\square$

*proof of Property 4.* (1) If $f(\mathrm{round}_S(A)) > F(A)$ then $\mathrm{argmin}_{k=1}^{M}(f(P_k)) > F(A)$ so for each $P_k$ in the decomposition $F(A) = \sum_{k=1}^{M} \alpha_k f(P_k)$ we have $f(P_k) > F(A)$. However, since $\sum_{k=1}^{M} \alpha_k = 1$ this implies $\sum_{k=1}^{M} \alpha_k f(P_k) > F(A)$, a contradiction.

(2) Let the decomposition of $F$ be $F(A) = \sum_{k=1}^{M} \alpha_k f(P_k)$. Recall by Thm. 2.7 that Alg. 2 produces this decomposition. Through Alg. 2 we have $P_1 = \mathrm{argmax}_{P \in \mathcal{P}_n} \langle P, S \rangle$. Next we show $\mathrm{argmax}_{P \in \mathcal{P}_n} \langle P, S \rangle = P^*$. Since each entry of $S$ is within $1/2n$ of $P^*$, we have

$$\langle S, P^* \rangle > n \left(1 - \frac{1}{2n}\right) \tag{11}$$

$$= n - \frac{1}{2}. \tag{12}$$

Also, any $P' \neq P^*$ must differ from $P^*$ by at least one entry so $\langle P', P^* \rangle \leq (n-1)$ and the inner product with $S$ is

$$\langle S, P' \rangle \leq (n-1) + n \left(\frac{1}{2n}\right) \tag{13}$$

$$= n - \frac{1}{2} \tag{14}$$

where the second term accounts for the entry-wise differences between $S$ and $P^*$. We have then shown $P_1 = \mathrm{argmax}_{P \in \mathcal{P}_n} \langle P, S \rangle = P^*$ which implies $f(\mathrm{round}_S(A)) \leq f(P^*)$. $\square$

*proof of Thm. 2.7.* Our algorithm for constructing the continuous Birkhoff decomposition is Alg. 2, which returns only the non-zero Birkhoff coefficients. This algorithm is the same as Alg. 1 except at each step, with $B$ being the matrix to be decomposed at this step, we subtract off the permutation of maximum score that is a matching of $B$, as opposed to an arbitrary matching of $B$. Recall that $B$ is proportional to a doubly stochastic matrix, so it either has a matching or is the zero matrix (in which case the algorithm terminates). To compute $\operatorname{argmax}_{P \in \mathcal{P}(B)} \langle P, S \rangle$ we first construct a bipartite graph $G$ that has an edge from vertex $i$ to vertex $j$ with weight $S(i,j)$ if and only if $B(i,j) > 0$. It is easy to see that (i) matchings of $G$ correspond to the matchings of the scaled doubly stochastic matrix $B$; and (ii) for any matching $P$ in this graph, its weight is exactly $\langle P, S \rangle$ which is the score of permutation $P$. Hence we can compute $\operatorname{argmax}_{P \in \mathcal{P}(B)} \langle P, S \rangle$ simply by computing the maximum-weight matching of this bipartite graph $G$. This computation takes $O(n^3)$ time using the Hungarian algorithm, and since there are at most $O(n^2)$ matchings to compute, the total time complexity is $O(n^5)$.

We show the correctness of Alg. 2. Recall our algorithm returns a collection of permutations $P_1, \ldots, P_M$. First, let $\Omega = \{\widehat{P}_\ell\}_{\ell=1}^{n!}$ denote the total ordering of all permutations induced by the score matrix $S$. Now let $A = \sum_{\ell=1}^{n!} \widehat{\alpha}_\ell \widehat{P}_\ell$ denote the Birkhoff decomposition of $A$ w.r.t. the total order $\Omega$ as defined in Def. 2.3. Note the slight change in notation so that $\hat{\alpha}_\ell$ and $\widehat{P}_\ell$ represent the permutations and coefficients in Def. 2.3 and $\alpha_k$ and $P_k$ represent the permutations and coefficients returned by Alg. 2. Let $i_1 < i_2 < \cdots < i_R$ denote the set of indices whose corresponding coefficients $\alpha_{i_k}$ are positive. That is, ignoring all zero coefficients in the decomposition, we have $A = \sum_{k=1}^{R} \widehat{\alpha}_{i_k} \widehat{P}_{i_k}$ where each $\widehat{\alpha}_{i_k} > 0$. Our goal is for each $k \in [M]$ to show (cond-A): that $\alpha_k = \widehat{\alpha}_{i_k}$ and $P_k = \widehat{P}_{i_k}$. We do so via induction on the index $k \in [M]$.

We begin by showing the property that $\widehat{P}_\ell$ is a matching of $B^{\ell-1} = A - \sum_{m=1}^{\ell-1} \widehat{\alpha}_m \widehat{P}_m$ if and only if $\widehat{\alpha}_\ell > 0$. Call this property $(*)$. This property holds since $\widehat{\alpha}_\ell > 0$ if and only if every element in the minimum defining $\widehat{\alpha}_\ell$ in Def. 2.3 is non-zero, which means for each $i, j \in [n]$ with $\widehat{P}_\ell(i,j) = 1$ we have $B^{\ell-1}(i,j) > 0$, i.e., $\widehat{P}_\ell$ is a matching of $B^{\ell-1}$.

The first permutation $P_1$ returned by Alg. 2 is $\operatorname{argmax}_{P \in \mathcal{P}(A)} \langle P, S \rangle$ which is the matching of $A$ with largest score. That is, $P_1$ is the matching of $A$ with smallest index in the total order $\Omega$. Furthermore, by $(*)$, the permutation $\widehat{P}_{i_1}$ must be a matching of $B^{i_1-1} = A$ since $\widehat{\alpha}_{i_1} > 0$. For $j < i_1$ we have $\widehat{\alpha}_j = 0$, and again by $(*)$, each $P_j$ is not a matching of $B^{j-1} = A$. We have shown $\widehat{P}_{i_1}$ is the matching of $A$ with smallest index in $\Omega$ and, therefore, (cond-A) holds for the base case $k = 1$.

Now assume (cond-A) holds for all $m < k$; we aim to show that it holds for $k$. Let $i_1 < i_2 < \cdots < i_{k-1}$ be the indices for the previous $k-1$ non-zero coefficients. In the $k$th iteration of Alg. 2, $B = A - \sum_{m=1}^{k-1} \alpha_m P_m = B^{i_{k-1}}$, and $P_k = \operatorname{argmax}_{P \in \mathcal{P}(B)} \langle P, S \rangle$, which is the first permutation matrix in the total order $\Omega$ that is a matching for $B$. Let $j$ be the index of $P_k$ in the total order $\Omega$, that is, $P_k = \widehat{P}_j$. Notice that $j \notin \{i_1, \ldots, i_{k-1}\}$ as this would contradict $P_k$ being a matching of $B$. We claim that $j > i_{k-1}$. If not, then by $(*)$, $\widehat{\alpha}_j > 0$, a contradiction to our inductive hypothesis that $i_1, \ldots, i_{k-1}$ are the first $k-1$ indices whose coefficients are non-zero in the Birkhoff decomposition of $A$. Hence, $P_j$ is the first permutation in the list $\widehat{P}_{i_{k-1}+1}, \ldots, \widehat{P}_{n!}$ that is a matching of $B$.

On the other hand, since $i_k$ is the first index in $i_{k-1}+1, i_{k-1}+2, \ldots, n!$ such that $\widehat{\alpha}_{i_k} > 0$, by $(*)$, the index $i_k$ is the first in this list such that the corresponding permutation is a matching of $B^{i_{k-1}-1} = B$. We can then conclude $i_k = j$ and $\widehat{P}_{i_k} = P_k$. Furthermore, $\hat{\alpha}_{i_k} = \alpha_k$ since $B = B^{i_{k-1}-1} = B^{i_k-1}$ so the definition of $\alpha_k$ in Alg. 2, $\alpha_k = \min_{ij}\{B(i,j) \mid P_k(i,j) = 1\}$ is equivalent to the definition of $\widehat{\alpha}_{i_k}$ in Eq. 3. This finishes the proof of the inductive step. Combining the base case with the inductive step, we have that (cond-A) holds for all $k \in [M]$, hence the set $\{(\alpha_k, P_k)\}_{k=1}^{M}$ returned by Alg. 2 exactly corresponds to those terms in the Birkhoff decomposition (as computed by Def. 2.3) with non-zero coefficients. □

# B  Unsupervised Neural optimizer

Since Birkhoff extensions are a.e. differentiable, they can be useful for training neural networks for unsupervised neural combinatorial optimization. In particular, similar to [29], we can train a neural network $N_\theta$ with parameters $\theta$ that maps an instance $I$ of a problem to a doubly stochastic matrix $A_I$,

which we aim to train for the optimization of the combinatorial objective $f$. See the illustration in Figure 1.

For example, for TSP in the Euclidean space $\mathbb{R}^d$ the instance is a set of cities, represented by a vector $X_I \in \mathbb{R}^{nd}$ representing $n$ points $\{x_1, \ldots, x_n\}$ in $\mathbb{R}^d$. The output of $N_\theta$ is a $n \times n$ doubly stochastic matrix $A_I = N_\theta(X_I)$, and the neural network is trained in an unsupervised manner to optimize $F(A_I)$. Note that once trained, when a new instance $I'$ is given with input $X_{I'}$, we can simply return $\text{round}_S(N_\theta(X_{I'}))$ as the TSP tour. Essentially, $N_\theta$ can be viewed as an neural optimizer for the given optimization problem over the extended space $\mathcal{D}_n$. Once a solution in $\mathcal{D}_n$ is identified, it can be rounded to a permutation without lowering the quality of the solution (Property 4).

Having a differentiable Birkhoff extension allows us to train such a neural network model in an unsupervised manner. In particular, first, suppose we have a score matrix $S$ – this score matrix can be chosen simply as a random stochastic matrix; or it can also be a canonical choice depending on the input problem instance. For example, in the case of TSP, we can choose $S$ to be a perturbation of the permutation derived from the MST. With this choice of $S$, let $A = N_\theta(X_I)$ be the output of the neural network. We have that (computed by Alg. 2)

$$F_S(A) = \sum_{k=1}^{M} \alpha_k(A) f(P_k(A)). \tag{15}$$

Here, note that both $\alpha_k$ and $P_k$ depend on $A$, and $A$ itself depends on the parameters $\theta$ of the neural network $N_\theta$. We simply minimize $F_S(A)$ w.r.t. the parameters $\theta$ via backpropagation. More precisely, computing $\frac{\partial F_S(A)}{\partial \theta}$ boils down to computing $\frac{\partial \alpha_k}{\partial \theta} = \frac{\partial \alpha_k}{\partial A} \cdot \frac{\partial A}{\partial \theta}$ for each positive Birkhoff coefficient $\alpha_k$.

The above description is for training $N_\theta$ only for a single instance. Usually one wishes to train $N_\theta$ over a family of instances $\mathcal{I}$, so that once trained, it can be used to produce solutions to new instances. In particular, during training, the loss is $\sum_{I \in \mathcal{I}} \frac{1}{|\mathcal{I}|} F_S(N_\theta(I))$. Once trained, given a new instance $I$, we can simply compute $A_I = N_\theta(X_I)$ and return the permutation $\text{round}_S(A_I)$ as a candidate solution. In practice, we found that for a test instance it makes sense to optimize $N_\theta$ for a few more iterations at testing (as a fine-tuning) to further improve the quality $A_I$.

We also note that for the case where we have a score matrix that depends on the problem instance (e.g, using MST to induce a score matrix for the TSP problem), it is beneficial to also take this score matrix $S_I$ as input to the neural network $N_\theta$ to better inform the output matrix $A_I = N_\theta(X_I, S_I)$. This input is optional.

Finally, in the simplest form, $N_\theta : \mathbb{R}^{nd} \to \mathbb{R}^{n \times n}$ maps an $n$-vector $X$ with each entry from $R^d$ to a $n \times n$ matrix $A$. In particular, again using TSP as an example, here we represent a set of points as a vector $X \in \mathbb{R}^{nd}$, which assumes an ordering of these points. The output matrix assumes the same ordering of input points. In other words, this map $N_\theta$ needs to be *permutation equivariant*, namely, if we permute the order of input points in $X$, then the output should permute in the same way. Mathematically, this means that $N_\theta$ satisfies $N_\theta(PX) = PN_\theta(X)P^T$ for any permutation matrix $P$ over $n$ elements. Such a permutation equivariant neural network can be implemented using models such as the set2graph neural network of [47] and the equivariant-graph network of [35].

We train a neural network $N_\theta$ for solving TSP using a loss comprised of the Birkhoff extension $F$ and a penalty term $\lambda \left( \sum_i \left( \sum_j A_{ij} - 1 \right)^2 + \sum_j \left( \sum_i A_{ij} - 1 \right)^2 \right)$ that ensures double stochasticity, where $\lambda$ is the weight. We use the Adam optimizer [1] to train the model. We generate a training dataset with $N = 6000$ instances and with a mixture of instance sizes, $n = 20, 30$, and $40$. The input to $N_\theta$, for each instance, is the vector $X_I$ and the MST-derived score matrix $S_t$. We trained $N_\theta$ for $T = 100$ epochs, and selected a learning rate of $\eta = 0.001$ using a hyperparameter search of the set $\{0.01, 0.05, 0.001, 0.0005\}$. At testing we optimize the trained neural network $N_\theta$ for a few more iterations as a fine-tuning. We compare the performance of this fine-tuned model (labeled *MST S Init. NN w. Alg. 2*) with the corresponding untrained model (labeled *MST S Init. Alg. 4*) in Table 4. We show that in most of the cases, the trained neural network model can achieve similar solution quality to the untrained model with much less runtime. This result holds even for problem instances that are larger than the instances seen in training

| Method | TSP: Tour Length↓ | | | | | | | | | |
|---|---|---|---|---|---|---|---|---|---|---|
| | **n = 20** | | **n = 30** | | **n = 40** | | **n = 50** | | **n = 100** | |
| | Obj. | Time | Obj. | Time | Obj. | Time | Obj. | Time | Obj. | Time |
| MST | 4.746 | < 1s | 5.784 | < 1s | 6.835 | < 1s | 7.410 | < 1s | 10.288 | < 1s |
| Random $S$ Init. Alg. 4 | 4.518 | 192s | 5.681 | 259s | 7.139 | 612s | 8.433 | 671s | 13.560 | 967s |
| MST $S$ Init. Alg. 4 (Improv.) | 4.345 | 156s | 5.290 | 214s | 6.328 | 571s | 6.892 | 640s | 9.836 | 1318s |
| MST $S$ Init. Alg. 4 (100 steps) | 4.541 | 2.71s | 5.512 | 5.30s | 6.529 | 8.29s | 7.333 | 9.72s | 10.239 | 24.56s |
| MST $S$ Init. NN w. Alg. 2 (Fine-tuned) | 4.180 | 1.75s | 5.206 | 2.15s | 6.219 | 2.59s | 7.170 | 3.02s | 10.220 | 5.42s |

Table 4: Performances of trained neural network and pure optimization both with Birkhoff extension on TSP.

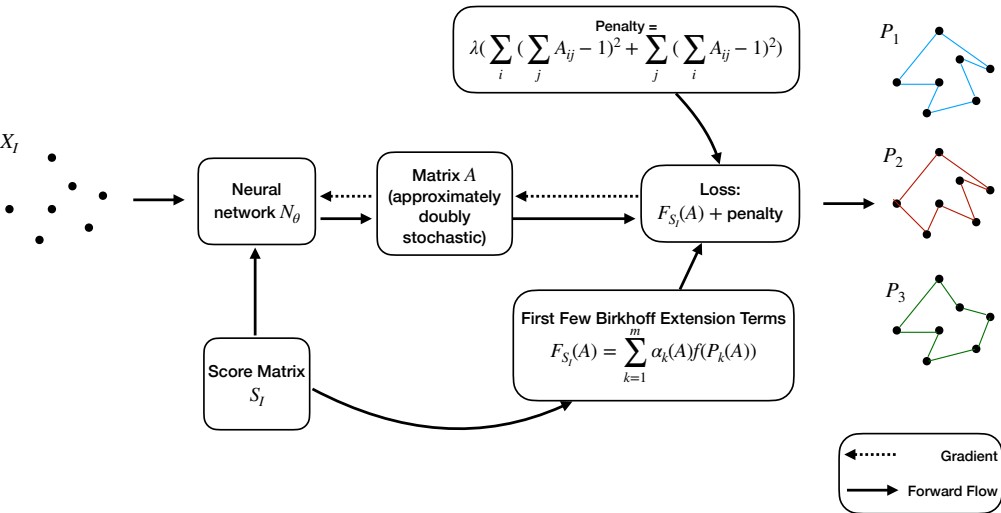

Figure 1: The pipeline of training a neural network $N_\theta$ for a single instance. For a given problem instance $I$, we have its representation $X_I$ and a score matrix $S_I$ as input to the neural network $N_\theta$. The output of the neural network is a doubly stochastic matrix $A = N_\theta(X_I, S_I)$. Birkhoff extensions are used to compute the loss $F_S(A) = \sum_{k=1}^{M} \alpha_k(A) f(P_k(A))$ and we minimize it via backpropagation. In the figure above, for example, $M = 3$ and rounding produces the permutation $P_2 = \text{round}_{S_I}(A_I)$, highlighted in red.

## C    Extensions for Optimization over Trees

In this section we consider how a parallel framework can be applied to optimization over rooted binary trees with $n$ labeled leaves. Problems such as the Steiner minimum tree problem and optimization-based hierarchical clustering reduce to optimization over this space. For instance, given a fixed rooted binary tree with leaves the set of terminals, the optimal Steiner tree with this topology can be computed efficiently. Therefore, computing the Steiner minimum tree reduces to optimizing the topology.

We begin by representing the space of rooted binary trees as matrices.

**Definition C.1.** *Let $\mathcal{T}_n$ be the space of rooted binary trees with leaf set $[n]$, which we represent by a directed graph with edges directed away from the root. Let $\mathcal{W}_{2n-2} \subset \mathcal{D}_{2n-2}$ be the space of $(2n-2) \times (2n-2)$ doubly stochastic matrices $W$ that satisfy $W(i,j) = 0$ if either*

1. *$i > n-1$ and $i \leq j$, or*

2. *$i \leq n-1$ and $i + (n-1) \leq j$.*

*Let $\mathcal{B}_{2n-2} \subset \mathcal{W}_{2n-2}$ be matrices in $\mathcal{W}_{2n-2}$ that have binary entries. We relate these spaces by the map*

$$\tau : \mathcal{B}_{2n-2} \to \mathcal{T}_n \tag{16}$$

$$B \mapsto T \tag{17}$$

*where $T$ is the tree with vertices $[2n-1]$, leaves $[n]$, and for $n-1 < i \leq 2n-1$ and $j < 2n-1$ the tree $T$ has an edge $(i,j)$ iff $B(i,j) = 1$ or $B(i-(n-1),j) = 1$.*

**Lemma C.2.** *$\tau$ is well-defined and surjective.*

*Proof.* First, we claim that for any $B \in \mathcal{B}_{2n-2}$ the image $\tau(B)$ is indeed a tree in $\mathcal{T}_n$. Note that since the columns of $B$ sum to 1, each vertex of $\tau(B)$ other than the root, which is vertex $2n-1$, has in-degree 1. Similarly, since the row sums of $B$ are 1, each of the internal vertices (non-leaf vertices), which are the vertices $\{n+1, \ldots, 2n-1\}$, has out-degree 2. (For each internal vertex, there are two rows in $B$ that give its children.) Furthermore, $\tau(B)$ has no edges $(i,j)$ where $i \leq j$ as the only entries in $B$ that can give rise to such edges are zero by (1.) and (2.) of Def. C.1. We can then conclude that $\tau(B)$ is a directed acyclic graph. Furthermore, $\tau(B)$ has no cycles (regardless of direction) since for such a cycle to not be a directed cycle it must contain a vertex with in-degree greater than 1.

For surjectivity, consider an arbitrary $T \in \mathcal{T}_n$. Let $\phi : \{n+1, \ldots, 2n-1\} \to \{n+1, \ldots, 2n-1\}$ be an enumeration of the internal vertices of $T$ that respects topological order, i.e., $\phi(i) > \phi(j)$ implies $j$ is not a descendent of $i$. For each internal vertex $i$ in $T$ with children $j, k$ let $B(i,j) = 1$ and $B(i+n-1, k) = 1$. Since $\phi$ respects the topological order, we can guarantee there are no entries $B(i,j) > 0$ with either $i \leq j$ and $i > n-1$ or $i + (n-1) \leq j$ and $i \leq n-1$, so (1.) and (2.) of Def. C.1 are satisfied and $B \in \mathcal{B}_{2n-2}$. This $B$ satisfies $\tau(B) = T$, thus, $\tau$ is surjective. $\square$

Matrices in the space $\mathcal{W}_{2n-2}$ can be decomposed using Birkhoff decomposition as $\mathcal{W}_{2n-2} \subset \mathcal{D}_{2n-2}$. Additionally, if $W \in \mathcal{W}_{2n-2}$ then its Birkhoff decomposition only contains permutations in $\mathcal{B}_{2n-2} \subset \mathcal{P}_{2n-2}$. By this fact, we are then free to apply Birkhoff extension to extend any function on $\mathcal{B}_{2n-2}$ to a function $F : \mathcal{W}_{2n-2} \to \mathbb{R}$, even if $f$ is not defined for $\mathcal{P}_{2n-1} \setminus \mathcal{B}_{2n-1}$.

Additionally, we can extend functions on trees $\mathcal{T}_n$ to functions on $\mathcal{W}_{2n-2}$; the procedure is as follows. First $f$ is composed with $\tau$ to yield a function $f \circ \tau : \mathcal{B}_{2n-2} \to \mathbb{R}$. Then Birkhoff extension is used to extend this function to $F : \mathcal{W}_{2n-2} \to \mathbb{R}$. If the extension $F$ is optimized to find some solution $W \in \mathcal{W}_{2n-2}$ the Birkhoff extension rounding scheme can be used to find a $B \in \mathcal{B}_{2n-2}$ such that $f(\tau(B)) \leq F(W)$. Here, $\tau(B)$ is a tree $T$ satisfying $f(T) \leq F(W)$, so this procedure can be used to optimize $f$.

## D    Problem Details

In this section we give more detailed problem definitions and show the integer linear programs we employ for optimization using Gurobi.

### D.1 Traveling Salesperson Problem

**Definition D.1.** *Given a set of $n$ cities $\{1, 2, \ldots, n\}$ and a distance $d_{i,j}$ between each pair of cities $i$ and $j$, the traveling salesperson problem (TSP) is to find the permutation $\pi : [n] \to [n]$ that minimizes*

$$\sum_{k=1}^{n} d_{\pi(k),\pi(k+1 \pmod n)} \tag{18}$$

TSP can be formulated as an integer linear program using subtour elimination constraints such as in the Miller-Tucker-Zemlin formulation [36]. For optimization with Gurobi we use the formulation given in [26].

Our experiments use the integer quadratic program $\min_{P \in \mathcal{P}_n} \sum_{i=1}^{n-1} P^T(i)DP(i+1) + P^T(n)DP(1)$ and its relaxation $\min_{P \in \mathcal{D}_n} \sum_{i=1}^{n-1} P^T(i)DP(i+1) + P^T(n)DP(1)$ for TSP. Here $P(i)$ denotes the $i$-th column of the square matrix $P$.

### D.2 DFASP

**Definition D.2.** *Let $G = (V, E)$ be a directed graph where $V$ is the set of vertices and $E$ is the set of directed edges. A feedback arc set in $G$ is a subset of edges $F \subseteq E$ such that the subgraph $G' = (V, E \setminus F)$ is acyclic. The Directed Feedback Arc Set Problem (DFASP) is to find the minimum feedback arc set, i.e., the smallest subset of edges $F$ whose removal makes the graph $G$ acyclic.*

Alternatively, DFASP can be formulated as a vertex ordering (i.e. permutation) problem, where the goal is to find an enumeration of vertices $\{v_i\}_{i=1}^{n}$ that minimizes the cardinality of the set of backward edges. Here we give an LP formulation. Let $x_{ij}$ be a binary variable that equals 1 if the edge $(i, j) \in E$ is a backward edge, and 0 otherwise. Let $y_i$ be an integer variable for each vertex $i \in V$ representing the position of vertex $i$ in a topological ordering. DFASP can be formulated using the following integer linear program of [**?** ].

$$
\begin{aligned}
\text{minimize} \quad & \sum_{(i,j)\in E} x_{ij} \\
\text{subject to} \quad & x_{ij} \in \{0, 1\} && \forall (i, j) \in E, \\
& y_i \in \mathbb{Z} && \forall i \in V, \\
& y_i + 1 \leq y_j + |V| \cdot x_{ij} && \forall (i, j) \in E
\end{aligned}
$$

### D.3 QAP

**Definition D.3.** *Given two non-negative $n \times n$ matrices $D$ and $L$ find a permutation $\pi : [n] \to [n]$ that minimizes*

$$\sum_{i,j \in [n]} D(i,j)L(\pi(i), \pi(j)). \tag{19}$$

We use the Kaufman-Broeckx integer linear program formulation [30] for the Gurobi implementation.

## E Experiments Details

In this section we provide additional experimental results for the TSP, DFASP and QAP.

### E.1 Optimization Curves

Here, we show the optimization curves for each experiment.

Below are plots of the averaged optimization curves for TSP on problem instances of different scales. The $x$-axis gives the number of steps $t$ and the $y$-axis gives solution tour length, averaged over the $N = 50$ instances.

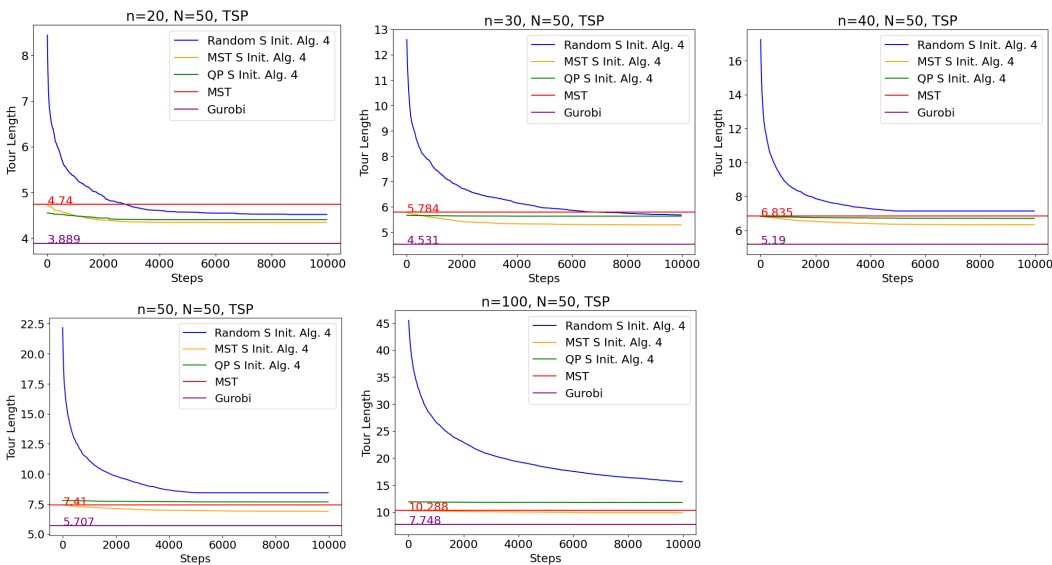

Below are plots of the optimization curves for DFASP. The $x$-axis gives the number of steps $t$ and the $y$-axis gives the average cardinality of the feedback arc set returned by the optimization algorithm. The runtime limit for both methods is $\frac{n}{10}$ minutes (e.g. 2 minutes for $n = 20$, 5 minutes for $n = 50$ and 10 minutes for $n = 100$). Results are averaged across $N = 50$ instances.

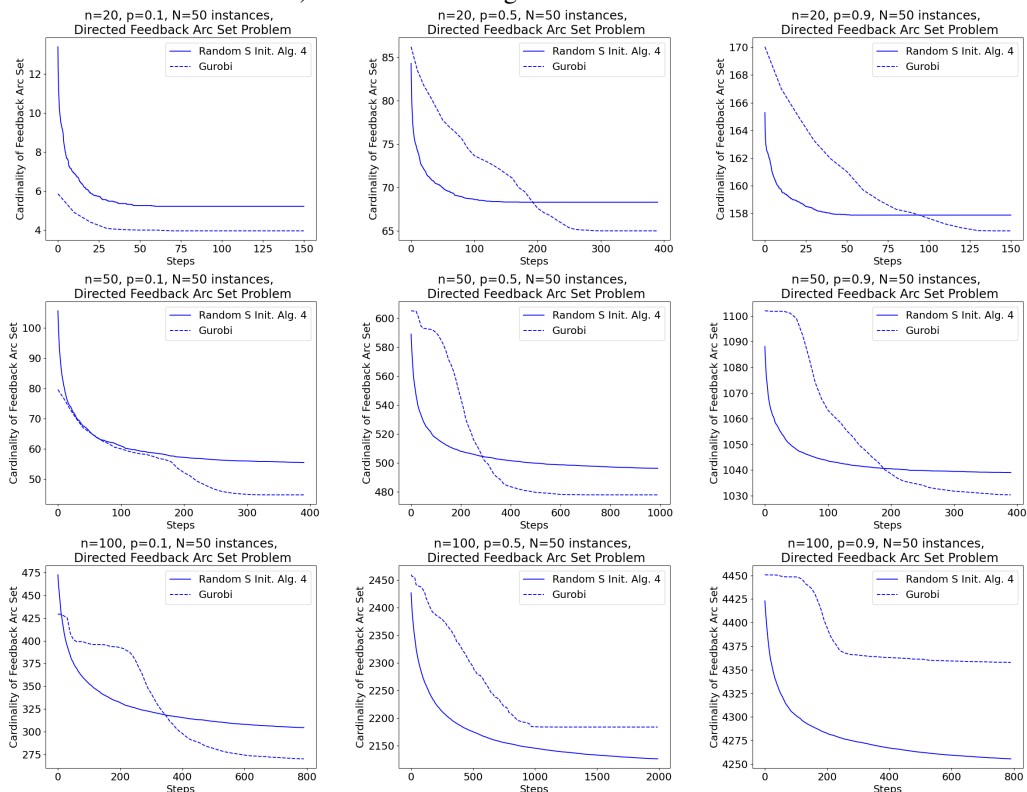

## E.2 Timing

We are using CentOS Linux 7 system on a Intel Xeon CPU E5-2650L processor. Gurobi is run on a single thread per process. We report average runtime for each method for different scales of TSP and QAP problems, see Table 5 and Table **??** below. DFASP is not included as we fix runtime for this problem

| Method | TSP | | | | |
|---|---|---|---|---|---|
| | **n = 20** | **n = 30** | **n = 40** | **n = 50** | **n = 100** |
| Gurobi | < 1s | < 1s | < 1s | < 1s | < 1s |
| MST | < 1s | < 1s | < 1s | < 1s | < 1s |
| QP | 36s | 50s | 69s | 95s | 240s |
| Random $S$ Init. Alg. 4 | 192s | 259s | 612s | 671s | 967s |
| MST $S$ Init. Alg. 4 | 156s | 214s | 571s | 640s | 1318s |
| QP $S$ Init. Alg. 4 | 135s | 249s | 532s | 615s | 953s |

Table 5: Average runtime of TSP algorithms in seconds.

## F  Ablation Study

This section provides additional ablation studies and extended experiments to support the main results. We examine the effects of truncation depth $k$, score-update frequency $m$, initialization strategy, optimization method, and instance size. Each subsection specifies the experimental setup used to isolate the respective factor.

### F.1  Effect of Truncation Depth $k$

To analyze the influence of the truncation depth $k$ in the Birkhoff decomposition, we conduct controlled experiments on a subset of 14 QAP instances selected from **QAPLIB**, with problem sizes ranging from $n = 12$ to $n = 35$. All experiments use identical hyperparameters: learning rate $\eta = 0.001$, total runtime $t = 2n$ seconds, and optimization via the Frank–Wolfe (FW) algorithm with random initialization. We vary $k \in \{3, 5, 10, 15, 20\}$ and report the average gap from the optimal objective. Figure 2 below shows that performance improves as $k$ increases up to $k{=}10$, then the performance decreases. This indicates that a moderate truncation depth provides a good balance between efficiency and solution quality.

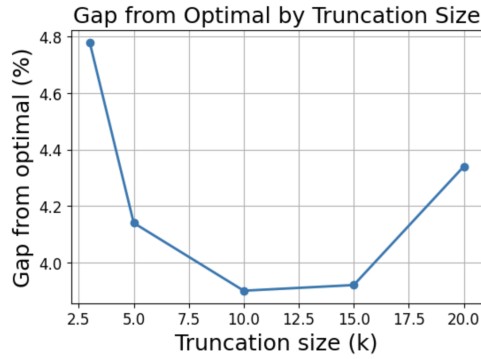

Figure 2: Gap vs. truncation size

Figure 3: Effect of score-update frequency

### F.2  Effect of Score-Update Frequency

We study the impact of the score-update frequency $m$, which controls how often the scoring matrix is recomputed during optimization. The same 14 QAPLIB instances and identical optimization settings are used as in the truncation experiments, fixing $k{=}5$. We test $m \in \{1, 5, 10, 20, 50\}$, where $m{=}1$ means the score matrix is refreshed at every iteration. Figure 3 show that more frequent updates consistently yield smaller optimality gaps, demonstrating that dynamic score re-weighting effectively helps the optimizer escape poor local minima.

## F.3 Initialization Strategy

We compare **random initialization** and **barycenter initialization** to understand their effects on optimization stability. Both configurations are evaluated on the same 14 QAPLIB instances with identical hyperparameters ($k$=5, $\eta$=0.001, runtime $t$=2$n$ seconds). In the barycenter case, optimization starts from the uniform doubly stochastic matrix, while the random case samples each entry uniformly from $[0, 1]$ before projecting onto the Birkhoff polytope via Sinkhorn iterations. Both use the same Frank–Wolfe optimizer. The comparison shows that random initialization achieves a slightly smaller average gap to optimal (4.14% vs. 4.42%), see results below in Table 6, suggesting that stochasticity in initialization provides beneficial diversity in the optimization trajectory.

| Initialization Method | Avg. Gap from Optimal (%) $\downarrow$ |
|---|:---:|
| Random Initialization | **4.14** |
| BaryCenter Initialization | 4.42 |

Table 6: Comparison of initialization strategies on 14 QAPLIB instances ($n = 12$–35). Both methods use identical hyperparameters ($k = 5$, $\eta = 0.001$, runtime 2$n$ s). Random initialization achieves a slightly lower average gap from optimal than barycenter initialization.

## F.4 Optimization Method Comparison

We compare two continuous optimization schemes over the Birkhoff polytope: (1) gradient descent with Sinkhorn projection and (2) the Frank-Wolfe (FW) algorithm proposed in this work. Both methods operate on the same 14 QAPLIB instances, with fixed $k$=5, learning rate $\eta$=0.001, and runtime $t$=2$n$ seconds. For the gradient-based variant, we apply 10 Sinkhorn iterations after each gradient step to enforce double stochasticity. FW achieves a lower average optimality gap (4.14%) than Gradient+Sinkhorn (4.55%), see results in Table 7 below.

| Optimization Method | Avg. Gap from Optimal (%) $\downarrow$ |
|---|:---:|
| Gradient + Sinkhorn Projection | 4.55 |
| Frank–Wolfe (ours) | **4.14** |

Table 7: Comparison between Frank–Wolfe and Gradient + Sinkhorn optimization over the Birkhoff polytope on 14 QAPLIB instances (sizes $n$=12–35). Frank–Wolfe achieves a lower average optimality gap. All experiments use identical hyperparameters ($k$=5, $\eta$=0.001, runtime 2$n$ s).

## F.5 Scalability on Large-Scale QAP Instances

To assess scalability, we perform large-scale experiments on **20 synthetic random QAP instances** of size $n$=1000. Because QAPLIB includes no instances larger than $n$=256, we generate our own dataset as follows: we uniformly sample 1000 facility points and 1000 location points in $[0, 1]^2$. The distance matrix $D$ is constructed from pairwise Euclidean distances among locations, and the flow matrix $F$ is defined as the inverse of pairwise distances among facilities. We evaluate BE using both Frank–Wolfe and Gradient+Sinkhorn optimizers under random and FAQ initialization. Each configuration is run with runtime limits of 2000 and 4000 seconds, corresponding to approximately 10,000 and 20,000 optimization epochs, respectively. We also include comparisons with Fast Approximate QAP (FAQ) and 2-opt baselines, see the results below in Table 8.

# G    Escaping Local Minima via Score Matrix Modification

In this section, we provide theoretical justification for Algorithm 4's strategy of modifying score matrices to escape local minima. We first establish a key lemma about the linearity of our Birkhoff extension along certain directions, then show that when the current solution is a local (but not global) minimum, there always exists a score matrix that allows escape from this local minimum.

**Lemma G.1** (Linearity Along Highest-Score Direction). *Let $f : \mathcal{P}_n \rightarrow \mathbb{R}$ be any function on permutations, and let $F_S$ be the Birkhoff extension of $f$ induced by score matrix $S$. Let $P$ be a permutation matrix that has the highest score under $S$. Then for any $A \in \mathcal{D}_n$ and $\beta \in [0, 1]$:*

$$F_S((1 - \beta)A + \beta P) = (1 - \beta)F_S(A) + \beta f(P)$$

| Method | Run-time (secs) | # of epochs | Avg. Objective ↓ | Gap from best % ↓ |
|---|---|---|---|---|
| Random assignment | — | — | 138,005,401 | 77.37% |
| 2-opt (Restart=3) | 2,000 | — | 119,216,266 | 53.13% |
| 2-opt (Restart=5) | 2,000 | — | 109,965,065 | 41.28% |
| FAQ | 345 | — | 77,894,862 | 0.10% |
| Ours (BE w/ Grad+Sink, Random Init.) | 2,000 | 10,133 | 100,894,812 | 29.64% |
| Ours (BE w/ Grad+Sink, Random Init.) | 4,000 | 20,064 | 90,244,871 | 15.97% |
| Ours (BE w/ FW, Random Init.) | 2,000 | 10,024 | 102,393,212 | 31.57% |
| Ours (BE w/ FW, Random Init.) | 4,000 | 20,091 | 89,114,150 | 14.40% |
| Ours (BE w/ FW, FAQ Init.) | 2,000 | 10,102 | 77,819,625 | **0.00%** |

Table 8: Comparison of QAP solution methods on large-scale instances.

*Proof.* Let $A = \sum_{k=1}^{M} \alpha_k P_k$ be the Birkhoff decomposition of $A$ under score matrix $S$. Consider $(1 - \beta)A + \beta P$.

We consider two cases:

Case 1: If $P \neq P_1$ (i.e., $P$ is not the first permutation in the decomposition of $A$), then the Birkhoff decomposition of $(1 - \beta)A + \beta P$ is simply $\beta P + (1 - \beta) \sum_{k=1}^{M} \alpha_k P_k$. This follows because:

- $P$ appears first with coefficient $\beta$ since it has highest score

- The remaining terms are proportional to $(1 - \beta)$ times the original decomposition

Therefore in this case:

$$F_S((1 - \beta)A + \beta P) = \beta f(P) + (1 - \beta)F_S(A)$$

Case 2: If $P = P_1$ (i.e., $P$ is the first permutation in the decomposition of $A$), then for any entry $(i, j)$ where $P(i, j) = 1$:

$$((1 - \beta)A + \beta P)(i, j) = (1 - \beta)(\alpha_1 + \sum_{k=2}^{M} \alpha_k P_k(i, j)) + \beta$$

$$= (1 - \beta)\alpha_1 + \beta + (1 - \beta)\sum_{k=2}^{M} \alpha_k P_k(i, j)$$

By definition of the Birkhoff decomposition, there exists at least one entry $(i^*, j^*)$ where $P(i^*, j^*) = 1$ such that $\sum_{k=2}^{M} \alpha_k P_k(i^*, j^*) = 0$ (this is what determines $\alpha_1$ in the original decomposition). For this entry:

$$((1 - \beta)A + \beta P)(i^*, j^*) = (1 - \beta)\alpha_1 + \beta$$

This means that in the Birkhoff decomposition of $(1 - \beta)A + \beta P$:

- $P$ must appear first (since it has highest score) with coefficient exactly $(1 - \beta)\alpha_1 + \beta$ (since this is the minimum value at any position where $P$ has a 1)

- After subtracting $((1 - \beta)\alpha_1 + \beta)P$, the remaining matrix is proportional to $(1 - \beta)$ times the matrix obtained after removing $\alpha_1 P$ from $A$'s decomposition

Therefore:

$$F_S((1-\beta)A + \beta P) = ((1-\beta)\alpha_1 + \beta)f(P) + (1-\beta)\sum_{k=2}^{M} \alpha_k f(P_k)$$

$$= ((1-\beta)\alpha_1)f(P) + \beta f(P) + (1-\beta)\sum_{k=2}^{M} \alpha_k f(P_k)$$

$$= (1-\beta)(\alpha_1 f(P) + \sum_{k=2}^{M} \alpha_k f(P_k)) + \beta f(P)$$

$$= (1-\beta)F_S(A) + \beta f(P)$$

In both cases we obtain $F_S((1-\beta)A + \beta P) = (1-\beta)F_S(A) + \beta f(P)$. □

*Proof of Thm 2.11.* We choose $S$ as $S = P^*$ where $P^*$ is some minimizer of $f$. This ensures that $P^*$ has strictly higher score than any other permutation under $S$. For any permutation matrix $P \neq P^*$, the score of $P^*$ under $S$ is $n$ (since each permutation has exactly $n$ 1's), while the score of $P$ under $S$ is equal to the number of positions where $P$ and $P^*$ agree, which is strictly less than $n$. Therefore, in the Birkhoff decomposition induced by $S$, $P^*$ appears as the first term.

Recall that a vector $v$ is called a descent direction at point $x$ if moving infinitesimally in direction $v$ decreases the function value. This can be verified by checking if the directional derivative is negative. For our extension $F\prime$, the directional derivative at $A$ in direction $P^* - A$ is:

$$\langle \nabla F_S(A), P^* - A \rangle = \lim_{\epsilon \to 0} \frac{F_S(A + \epsilon(P^* - A)) - F_S(A)}{\epsilon}$$

$$= \lim_{\epsilon \to 0} \frac{F_S((1-\epsilon)A + \epsilon P^*) - F_{S\prime}(A)}{\epsilon}$$

$$= \lim_{\epsilon \to 0} \frac{(1-\epsilon)F_S(A) + \epsilon f(P^*) - F_S(A)}{\epsilon} \qquad \text{(by Lem. G.1)}$$

$$= \lim_{\epsilon \to 0} \frac{\epsilon(f(P^*) - F_S(A))}{\epsilon}$$

$$= f(P^*) - F_S(A).$$

The third equality holds by Lem. G.1 since our construction of $S$ ensures $P^*$ appears first in any Birkhoff decomposition.

If $f(P^*) - F_S(A) = 0$ then $A$ must be a global minimum of $F_S$. If $f(P^*) - F_S(A) < 0$ then $P^* - A$ is a descent direction for $F_S$ at $A$. In either case, $A$ is not a local minimum of $F_S$.

□

This theorem provides theoretical justification for Algorithm 4's strategy of modifying score matrices when stuck at local minima. It shows that as long as better solutions exist (i.e., permutations with lower objective value), there always exists a score matrix that enables escape from the current local minimum.

## H   Comparison with Convex Closure

In this section, we compare our Birkhoff extension with the convex closure, a well-known theoretical concept in convex analysis. While the convex closure provides certain theoretical guarantees, our Birkhoff extension offers practical computability, making it more suitable for real-world optimization problems.

### H.1   Definition and Properties of Convex Closure

The convex closure of a function $f : \mathcal{P}_n \to \mathbb{R}$ is defined as the greatest convex function that is pointwise less than or equal to $f$ on the domain of permutation matrices. For our purposes, since we are extending $f$ from $\mathcal{P}_n$ to $\mathcal{D}_n$, the convex closure can be expressed as:

**Definition H.1** (Convex Closure). *Given a function $f : \mathcal{P}_n \to \mathbb{R}$, its convex closure $Conv(f) : \mathcal{D}_n \to \mathbb{R}$ is defined as:*

$$Conv(f)(X) = \inf \left\{ \sum_{i=1}^{k} \alpha_i f(P_i) \mid X = \sum_{i=1}^{k} \alpha_i P_i, \alpha_i \geq 0, \sum_{i=1}^{k} \alpha_i = 1 \right\}$$

*where the infimum is taken over all possible Birkhoff decompositions of $X$.*

The convex closure is convex by definition, agrees with $f$ on all permutation matrices, and for minimization problems, minimizing $Conv(f)$ over $\mathcal{D}_n$ gives the same optimal value as minimizing $f$ over $\mathcal{P}_n$.

## H.2 Computational Challenges and Advantages of Our Approach

Despite its theoretical appeal, computing the convex closure for a general function $f$ and doubly stochastic matrix $X$ is computationally intractable. Determining the convex closure requires solving a combinatorial optimization problem to find the optimal set of permutation matrices and their weights, which is generally NP-hard for large $n$.

In contrast, our score-based Birkhoff extension can be computed in $O(n^5)$ time for any doubly stochastic matrix. It is continuous and almost everywhere differentiable, enabling the use of gradient-based optimization methods. While not necessarily convex, our extension preserves the property that minimizing over $\mathcal{D}_n$ and rounding gives the same value as minimizing directly over $\mathcal{P}_n$ (Property 3). The flexibility provided by different score matrices allows for exploration of the solution space and escape from local minima.

The computational efficiency of our Birkhoff extension makes it applicable to practical problems where the convex closure would be intractable to compute. For optimization problems over permutations, we can use gradient-based methods efficiently, integrate with machine learning approaches like neural networks, and handle larger problem instances than would be possible with exact convex closure computations.

In summary, while the convex closure provides a theoretically optimal convex relaxation, our Birkhoff extension offers a practical, computable alternative that preserves many desirable properties while enabling efficient optimization.

# I  Generalization to Constrained Permutations

The framework presented in this paper can be naturally extended to consider only a subset of permutations that satisfy certain constraints. This generalization is particularly useful in applications where not all permutations are valid or desirable solutions. The key insight is that our framework can work with any subset of permutations for which we can efficiently solve the corresponding matching problem under the given constraints.

## I.1  Formal Setup

Let $\mathcal{C}_n \subseteq \mathcal{P}_n$ be a subset of permutation matrices that satisfy some constraints. We define the *constrained Birkhoff polytope $\mathcal{D}_n(\mathcal{C})$* as the convex hull of $\mathcal{C}_n$:

$$\mathcal{D}_n(\mathcal{C}_n) = \left\{ A \in \mathbb{R}^{n \times n} \mid A = \sum_{P \in \mathcal{C}_n} \alpha_P P, \sum_P \alpha_P = 1, \alpha_P \geq 0 \right\} \qquad (20)$$

For this framework to be practical, we require two key properties of the constraint set $\mathcal{C}_n$:

1. **Efficient Matching**: We must be able to efficiently solve the constrained matching problem $\operatorname{argmax}_{P \in \mathcal{C}_n} \langle P, S \rangle$ for any score matrix $S$. This same matching problem is used in two contexts in our framework: (1) finding the highest scoring permutation in the decomposition algorithm, and (2) finding the permutation with greatest inner product with the gradient in the Frank-Wolfe optimization step. The matching problem must be modified to respect the given constraints while still finding the permutation that maximizes the inner product.

2. **Efficient Initialization**: We must be able to efficiently generate points within the constrained polytope $\mathcal{D}_n(\mathcal{C})$. This is crucial because optimization methods like Frank-Wolfe require feasible starting points. For many constraint types, finding such initial points is straightforward.

The continuous decomposition framework can be adapted to work with $\mathcal{D}_n(\mathcal{C}_n)$ as follows:

---

**Algorithm 5** Constrained Continuous Birkhoff decomposition

---

**Require:** $A \in \mathcal{D}_n(\mathcal{C})$, identifying score matrix $S$
**Ensure:** $\{(\alpha_k, P_k)\}_{k=1}^M$ s.t. $A = \sum_{k=1}^M \alpha_k P_k$,
$\sum_k \alpha_k = 1$, $\alpha_k > 0$, and $P_k \in \mathcal{C}_n$.

> $k \leftarrow 1, B \leftarrow A$
> **while** $B \neq 0$ **do**
> > $P_k \leftarrow \mathrm{argmax}_{P \in \mathcal{C}_n} \langle P, S \rangle$
> > $\alpha_k \leftarrow \min_{ij}\{B(i,j) \mid P_k(i,j) = 1\}$
> > $B \leftarrow B - \alpha_k P_k$
> > $k{+}{+}$
> **end while**
> $M \leftarrow k$
> **return** $\{(\alpha_k, P_k)\}_{k=1}^M$

---

For optimization over the constrained polytope, we use the Frank-Wolfe algorithm with constrained matching:

---

**Algorithm 6** Constrained Frank-Wolfe over $\mathcal{D}_n(\mathcal{C})$

---

**Require:** Initial point $A_0 \in \mathcal{D}_n(\mathcal{C})$, score matrix $S$, step sizes $\{\gamma_t\}_{t=1}^T$
**Ensure:** Final doubly stochastic matrix $A_T \in \mathcal{D}_n(\mathcal{C})$
> $A_0 \leftarrow \mathrm{initialize}(\mathcal{D}_n(\mathcal{C}))$
> **for** $t = 1$ to $T$ **do**
> > $P_t \leftarrow \mathrm{argmax}_{P \in \mathcal{C}_n} \langle P, \nabla F_S(A_{t-1}) \rangle$
> > $A_t \leftarrow (1 - \gamma_t)A_{t-1} + \gamma_t P_t$
> **end for**
> **return** $A_T$

---

## I.2 Key Properties

The constrained version maintains two key properties of the original framework:

1. **Continuity**: The decomposition remains continuous as long as the constraint set $\mathcal{C}_n$ is fixed and the matching problem under constraints can be solved efficiently. This follows from the same argument as in the unconstrained case, where we fix an ordering over the constrained permutations. The continuity of the decomposition implies that the extension $F(A) = \sum_k \alpha_k f(P_k)$ of any function $f : \mathcal{C}_n \to \mathbb{R}$ is also continuous.
2. **Rounding**: The constrained version inherits all the rounding properties proved for the original Birkhoff extension, as the the extension is still a convex combination of permutations.

## I.3 Compatible Constraint Types

The framework can handle any constraint set $\mathcal{C}_n$ for which the matching problem $\mathrm{argmax}_{P \in \mathcal{C}_n} \langle P, S \rangle$ can be solved efficiently. A key class of constraints that satisfy this requirement are those whose linear programming relaxation has integral solutions. This means that when we relax the permutation constraints to allow for doubly stochastic matrices, the optimal solution of the constrained matching problem is still a permutation matrix. This property ensures that the matching problem can be solved efficiently using standard linear programming algorithms, and the solution will automatically satisfy the permutation constraints.

Most notably this includes LPs with a unimodular constraint matrix and an integer right-hand sides [45]. The integrality of the LP relaxation is a powerful property that guarantees the existence of efficient algorithms for solving the matching problem, making these constraints particularly well-suited for our framework.

A simple but important example is the case of "don't match" constraints, where certain pairs of elements are forbidden from being matched. This can be expressed by setting the corresponding entries in the score matrix $S$ to $-\infty$, effectively making these matches impossible. The LP relaxation of this constrained matching problem has integral solutions, as the optimal solution will never select a forbidden match. This basic case demonstrates how the framework can handle even simple constraints while maintaining its key properties.

## J   Truncated Birkhoff Extension

In practice, computing the full $S$-induced Birkhoff decomposition can be expensive. We introduce a truncated and normalized variant that uses only the first $K$ non-zero terms of the decomposition while preserving the convex-combination interpretation.

**Definition J.1** (Truncated, normalized Birkhoff extension). *Let $S$ be an identifying score matrix and let $(\alpha_k(A), P_k(A))_{k=1}^{M}$ be the non-zero terms in the $S$-induced Birkhoff decomposition of $A \in \mathcal{D}_n$, ordered as in Alg. 2. For any $K \in \{1, \ldots, M\}$, define the truncated normalization factor*

$$Z_K(A) = \sum_{k=1}^{K} \alpha_k(A), \tag{21}$$

*and the normalized coefficients*

$$\tilde{\alpha}_k^{(K)}(A) = \begin{cases} \dfrac{\alpha_k(A)}{Z_K(A)} & \text{if } k \leq K, \\ 0 & \text{if } k > K. \end{cases} \tag{22}$$

*The truncated Birkhoff extension of $f : \mathcal{P}_n \to \mathbb{R}$ at level $K$ is*

$$F_S^{(K)}(A) = \sum_{k=1}^{K} \tilde{\alpha}_k^{(K)}(A)\, f\big(P_k(A)\big), \tag{23}$$

*which satisfies $\sum_{k=1}^{K} \tilde{\alpha}_k^{(K)}(A) = 1$ by construction.*

**Theorem J.2** (Escaping Local Minima for Truncated Extension). *Let $f : \mathcal{P}_n \to \mathbb{R}$ and let $F_S^{(K)}$ be the truncated, normalized Birkhoff extension at level $K > 1$. There exists a score matrix $S$ such that any $A \in \mathcal{D}_n$ is not a local minimum of $F_S^{(K)}$.*

*Proof.* The proof parallels that of Thm. 2.11. Choose an identifying score matrix $S$ and choose it so that some $P^* \in \arg\min_P f(P)$ is the unique top-scoring permutation under $S$. Let the $S$-ordered decomposition of $A$ be $A = \sum_{i \geq 1} \alpha_i P_i$. Define the path $A(\beta) = (1 - \beta)A + \beta P^*$, $\beta \in [0, 1]$.

Let $J$ be the set of the top-$K$ permutations (by $S$) at $\beta = 0$ *after* inserting $P^*$ if needed: if $P^* \notin \{P_1, \ldots, P_K\}$ then $J = \{P^*\} \cup \{P_1, \ldots, P_{K-1}\}$; otherwise $J = \{P_1, \ldots, P_K\}$. Write

$$S_J = \sum_{P \in J} \alpha_P, \qquad T_J = \sum_{P \in J} \alpha_P\, f(P), \qquad \bar{f}_J(A) = T_J / S_J.$$

Since $S$ is identifying, the $S$-order is strict; hence for all sufficiently small $\beta > 0$, the top-$K$ set along $A(\beta)$ remains $J$ and all coefficients of terms in $J$ are scaled by $(1 - \beta)$ while $P^*$ gains $+\beta$. Therefore

$$F_S^{(K)}\big(A(\beta)\big) = \frac{(1 - \beta)\, T_J + \beta\, f(P^*)}{(1 - \beta)\, S_J + \beta}.$$

Differentiating at $\beta = 0^+$ gives

$$\frac{d}{d\beta} F_S^{(K)}\big(A(\beta)\big)\bigg|_{\beta=0} = \frac{f(P^*) - \bar{f}_J(A)}{S_J} \leq 0,$$

since $f(P^*) = \min_P f(P) \leq \bar{f}_J(A)$. If the inequality is strict, $P^* - A$ is a descent direction and $A$ is not a local minimum. If equality holds, then $\bar{f}_J(A) = f(P^*)$, so every $P \in J$ already attains the global value and $F_S^{(K)}(A) = f(P^*)$, i.e., $A$ already achieves the global minimum of $F_S^{(K)}$. This proves the claim for $K \geq 2$. $\qquad\square$

**Remark J.3** (Truncated preservation of properties). *All properties established for score-induced Birkhoff extensions continue to hold for the truncated, normalized extension $F_S^{(K)}$:*

- ***Efficient computation (Prop. 2).*** *Running Alg. 2 for $K$ iterations yields the truncated decomposition and $F_S^{(K)}$ in $O(K\,n^3)$ time (via $K$ maximum-weight matchings), which is $\leq O(n^5)$ as in the full case.*
- ***Minima correspondence (Prop. 3).*** *For any permutation $P$, $F_S^{(K)}(P) = f(P)$. For any $A$, $F_S^{(K)}(A)$ is a convex combination of $\{f(P_k(A))\}_{k \leq K}$, so $F_S^{(K)}(A) \geq \min_{P \in \mathcal{P}_n} f(P)$. Thus $\min_{A \in \mathcal{D}_n} F_S^{(K)}(A) = \min_{P \in \mathcal{P}_n} f(P)$, and any minimizer belongs to $\mathrm{Conv}(\mathrm{argmin}_{P \in \mathcal{P}_n} f(P))$ by the same argument as Prop. 3.*
- ***Rounding and approximation (Prop. 4).*** *Using the $K$ permutations returned by the truncated decomposition, define $\mathrm{round}_S^{(K)}(A) = \mathrm{argmin}_{k \leq K} f(P_k(A))$. Since $F_S^{(K)}(A)$ is an average over $\{f(P_k(A))\}_{k \leq K}$, we have $f(\mathrm{round}_S^{(K)}(A)) \leq F_S^{(K)}(A)$. The $C$-approximation transfer follows identically. For the score-update guarantee (Prop. 4–2), if $S'$ is sufficiently close to $P^*$, then $P^*$ is the top-scoring permutation under $S'$ and thus appears as the first term, so in particular among the first $K$ terms for any $K \geq 1$; the same argument goes through.*
- ***Dynamic score and escaping local minima.*** *Thm. J.2 provides the truncated analogue of Thm. 2.11; the proof follows the same linearity-along-$P^*$ direction combined with normalization.*

## K   Global Lipschitz bound for Birkhoff extension

We now provide an explicit Lipschitz bound for the (non-truncated) score-induced Birkhoff extension that is independent of the choice of score matrix.

**Theorem K.1** (Score-matrix–independent Lipschitz bound). *Let $f : \mathcal{P}_n \to \mathbb{R}$ and define the oscillation $\Delta_f = \max_P f(P) - \min_P f(P)$. For any identifying score matrix $S$ and any $A, A' \in \mathcal{D}_n$,*

$$|F_S(A) - F_S(A')| \leq \tfrac{\Delta_f}{2}\,\|A - A'\|_1. \tag{24}$$

*In particular, the Lipschitz constant in (24) is independent of $S$.*

*Proof.* By Prop. 1, $F_S$ is a.e. differentiable on $\mathcal{D}_n$ with

$$\nabla_A F_S(A) = \sum_{\ell \in L_+} (\nabla_A \alpha_\ell(A))\, f(P_\ell), \quad L_+ = \{\ell : \alpha_\ell(A) > 0\}.$$

From Def. 2.3, $\alpha_\ell(A)$ is the minimum over the $n$ entries on the support of $P_\ell$ after subtracting earlier terms. Hence, at points of differentiability, $\nabla_A \alpha_\ell$ is supported on a single entry (the active minimum), and different $\ell$ have disjoint supports along the decomposition path. Moreover, since $\sum_\ell \alpha_\ell(A) = 1$ for all $A$, we have $\sum_\ell \nabla_A \alpha_\ell(A) = 0$ entrywise. Indeed, the map $A \mapsto \sum_\ell \alpha_\ell(A)$ is the constant function 1, so its directional derivative in any direction $H$ is zero: $0 = \sum_\ell \langle \nabla_A \alpha_\ell(A), H \rangle$. As this holds for every $H$, it follows that the matrix $\sum_\ell \nabla_A \alpha_\ell(A)$ must be identically zero. Using this, for any scalar $c$,

$$\nabla_A F_S(A) = \sum_{\ell \in L_+} \nabla_A \alpha_\ell(A)\,(f(P_\ell) - c).$$

Choosing $c = \frac{\max_P f(P) + \min_P f(P)}{2}$ centers the values so that $|f(P_\ell) - c| \leq \Delta_f/2$ for all $\ell$. Because each entry of $\nabla_A F_S(A)$ receives contribution from at most one $\ell$ (disjoint supports), it follows that

$$\|\nabla_A F_S(A)\|_\infty \leq \tfrac{\Delta_f}{2}.$$

Along the segment $A_t = (1 - t)A + tA'$, the fundamental theorem of calculus gives

$$F_S(A') - F_S(A) = \int_0^1 \langle \nabla_A F_S(A_t),\, A' - A \rangle\, dt,$$

where $\langle \cdot, \cdot \rangle$ denotes the Frobenius inner product. Then $|\langle X, Y \rangle| \leq \|X\|_\infty \|Y\|_1$ for matrices $X, Y$ yields

$$|F_S(A') - F_S(A)| \leq \int_0^1 \|\nabla_A F_S(A_t)\|_\infty \, dt \, \|A' - A\|_1 \leq \tfrac{\Delta_f}{2} \|A' - A\|_1,$$

where we have applied the uniform bound on $\|\nabla_A F_S\|_\infty$ a.e. along the segment. $\qquad\square$

## L  GPU friendly maximum perfect matching

We present a suboptimal randomized algorithm for the Maximum Weight Perfect Matching problem in bipartite graphs, with a runtime of $O(n^3)$. While this problem can be solved deterministically in $O(n^3)$ time, and more efficiently in $O(n^\omega)$ time using randomized algorithms with low failure probability [37, 44], our goal is different: we aim to design an algorithm that is simple to implement and efficient on a GPU. Our algorithm runs in $O(n^3)$ time and returns a perfect matching with weight at least half of the optimal.

Let $G = (U, V, E)$ be a bipartite graph, where $|U| = |V| = n$, $U = \{u_1, ..., u_n\}$, $V = \{v_1, ..., v_n\}$. Let $A(G)_{i,j}$ be a random number from $\{1, ..., R\}$ if $(u_i, v_j) \in E$, otherwise $A(G)_{i,j} = 0$. Here $R$ is chosen to be large enough e.g., $R = n^{O(1)}$ and the matrix $A$ is known as random adjacency matrix of $G$. The matrix $A(G)$ has nice properties some of which we use here to design our algorithm. First the rank of $A(G)$ is at most equal to the size of the maximum matching and the equality holds with probability at least $1 - n/R$. More importantly, if $G$ has a perfect matching then with high probability $\det(A(G)) \neq 0$, and so $A(G)$ is invertible. Rabin and Vazirani [41] showed that

**Theorem L.1.** *With high probability, $A(G)_{i,j}^{-1} \neq 0$ if and only if graph $G - \{u_i, v_j\}$ has a perfect matching.*

In particular for edge $e = (u_i, v_j)$, $A(G)_{i,j}^{-1} \neq 0$ if and only if $e$ is *allowed*, i.e. is contained in a perfect matching. We use this result and the techniques developed by [37] to design our greedy algorithm for finding a maximum weight perfect matching. We note that in the worst case our algorithm returns a perfect matching with weight at least 1/2 of the optimal one.

---

**Algorithm 7** 2-approximation maximum weight perfect matching

---

**Require:** Bipartite graph $G$
    $B = A(G)^{-1}$
    $M = \emptyset$
    **for** $c = 1 \cdots n$ **do**
        Find maximum weight edge $(u_c, v_r)$ that is an allowed edge in $G - M$ i.e. row $r$, not yet eliminated, and such that $B_{r,c} \neq 0$ and $A(G)_{c,r} \neq 0$.
        eliminate the $r$-th row and the $c$-th column of $B$
        Add $(u_c, v_r)$ to $M$
    **end for**
    **return** $M$

---

**Theorem L.2.** *Algorithm 7 runs in time $O(n^3)$ and with high probability returns a perfect matching $M$ such that the weight of $M$ is at least 1/2 times the weight of the optimal matching.*

*Proof.* The proof that Algorithm 7 runs in time $O(n^3)$ and with high probability returns a perfect matching follows from [37]. The approximation guarantee follows from our greedy selection among the feasible edges at each iteration. $\qquad\square$

