# OpenReview forum: "Differentiable extensions with rounding guarantees for combinatorial optimization over permutations"
_NeurIPS.cc/2025/Conference — NeurIPS 2025 poster_

### Official Review · Reviewer_TNph · 2025-06-26

**Clarity:** 3
**Significance:** 2
**Originality:** 4
**Rating:** 4
**Confidence:** 5

**Summary:**

The authors tackle optimization problems over the space of permutations.

They introduce **Birkhoff Extension (BE)**, a continuous, almost-everywhere differentiable relaxation of permutation functions defined on the Birkhoff polytope (the convex hull of permutation matrices).

They propose a scoring-matrix-based method for rounding.

They show good theoretical properties about rounding.

They also design optimization techniques, e.g., Frank-Wolfe for preserving doubly-stochasticness, and updating the scoring matrix for escaping from local minima.

On benchmark instances of QAP (from QAPLIB), random-Euclidean TSP, and Directed Feedback Arc Set (on random graphs), BE matches or outperforms Gurobi, classical heuristics, and existing approximation algorithms.

**Questions:**

- [Q1] Property 4-2 tells that the rounding result cannot be worse than the scoring matrix, but I am wondering whether the rounding can actually get better than the scoring matrix when S is “sufficiently close to some permutation P”.
- [Q2] Theorem 2.11 only tells us the existence of such an S, but I am wondering whether we can have any results on, e.g., how easily we can find such an S.

**Ethical Concerns:**

["NO or VERY MINOR ethics concerns only"]

**Final Justification:**

The authors tackle optimization problems over the space of permutations.

They introduce **Birkhoff Extension (BE)**, a continuous, almost-everywhere differentiable relaxation of permutation functions defined on the Birkhoff polytope (the convex hull of permutation matrices).

They propose a scoring-matrix-based method for rounding.

They show good theoretical properties about rounding.

They also design optimization techniques, e.g., Frank-Wolfe for preserving doubly-stochasticness, and updating the scoring matrix for escaping from local minima.

On benchmark instances of QAP (from QAPLIB), random-Euclidean TSP, and Directed Feedback Arc Set (on random graphs), BE matches or outperforms Gurobi, classical heuristics, and existing approximation algorithms.

# Strengths

- **[S1] Important Problem**: Optimization problems on permutations are very important with both theoretical and empirical values.
- **[S2] Novel Differentiable Birkhoff Decomposition:** Although the classical Birkhoff Decomposition has been well known for a long time, the proposed differentiable Birkhoff decomposition based on the scoring matrix is novel and interesting.
- **[S3] Experiments with Broad Applicability: The proposed framework works across a variety of problems.**

**Limitations:**

Yes. Limitations are discussed in Section 4, “Concluding Remarks”.

**Quality:**

2

**Strengths And Weaknesses:**

# Strengths

- **[S1] Important Problem**: Optimization problems on permutations are very important with both theoretical and empirical values.
- **[S2] Novel Differentiable Birkhoff Decomposition:** Although the classical Birkhoff Decomposition has been well known for a long time, the proposed differentiable Birkhoff decomposition based on the scoring matrix is novel and interesting.
- **[S3] Experiments with Broad Applicability: The proposed framework works across a variety of problems.**

# Weaknesses

- **[W1] High Time Complexity**: O(n^5) time complexity can be prohibitive for large-scale problems. Also, Frank-Wolfe and dynamic score bring additional overheads.
- **[W2] Probably Not-Very-Practically-Meaningful Theory:**
    - See [Q1] and [Q2] below
- **[W3] Missing References:**
    - Min et al. "Unsupervised learning for solving the travelling salesman problem." NeurIPS’23
        - This work also discusses unsupervised learning for TSP
    - Bu et al. "Tackling prevalent conditions in unsupervised combinatorial optimization: Cardinality, minimum, covering, and more." ICML’24
        - This work also discusses theoretical properties of objective extension and rounding in unsupervised combinatorial optimization, as well as continuous relaxation for problems on trees

---

> ### Author Rebuttal · Authors · 2025-07-31
>
> We greatly appreciate the reviewers insightful comments and suggestions. To begin, we are thankful for the suggested references in W3, and will add these to the manuscript.
>
>
> Regarding Q1, if the score matrix is close to a permutation $P^\*$, then the rounding scheme can indeed produce a solution better than $P^\*$. All that is required for this to happen is that some permutation in the decomposition is better than $P^\*$. While Property 4-2 gives that  at least one permutation will be $P^\*$, there is nothing preventing the decomposition from containing better permutations.
>
> Theorem 2.11 shows the existence of an $S$ that allows for escaping a local minimum. However, the proof is not constructive, as, indeed, the score $S$ used in the proof is a permutation minimizing the permutation function $f$. Thus, the hardness of finding this $S$ is the same as optimizing $f$. However, as suggested in Q2, an interesting avenue for future research would be to investigate whether there are score matrices with desirable theoretical properties that are not computationally hard to find. For now, Theorem 2.11 only shows that the approach of changing $S$ always has the potential to escape local minima but does not provide any procedure for finding such an $S$.

---

> > ### Comment · Reviewer_TNph · 2025-08-05
> >
> > Thanks for the reply!
> > I think we are on the same page.
> > Although the theorems may not be that practical, I still believe this work is novel and impactful.
> > I keep my positive evaluation.
> > All the best to the authors.

---

### Official Review · Reviewer_s7f4 · 2025-06-30

**Clarity:** 2
**Significance:** 3
**Originality:** 3
**Rating:** 5
**Confidence:** 3

**Summary:**

The paper presents Birkhoff Extension (BE), an almost-everywhere differentiable continuous extension that lifts any function defined on permutation matrices to the Birkhoff polytope. The key idea is a score-induced Birkhoff decomposition, which selects permutations in a fixed order determined by a score matrix. The proposed method exhibits several appealing properties, such as minima correspondence and a rounding guarantee. The authors further develop a Frank-Wolfe optimizer with optional dynamic score updates to help escape local minima, and they apply this framework to unsupervised neural combinatorial optimization. Experimental results on QAP, TSP, and DFASP tasks demonstrate the effectiveness of the proposed approach.

**Questions:**

1. How does the number of decomposition terms $k$ affect solution quality? A theoretical bound or empirical curve would be helpful.

2. Can the method scale to larger problems (e.g., $n=1000$) with manageable runtime?

3. Can you provide sensitivity analysis for the key parameters like truncation depth and score update frequency?

4. How does this method compare to direct optimization over the Birkhoff polytope (e.g., Sinkhorn, PGD)?

**Ethical Concerns:**

["NO or VERY MINOR ethics concerns only"]

**Final Justification:**

This rebuttal address most of my concerns, and I decide to raise my rating.

**Limitations:**

yes

**Paper Formatting Concerns:**

Line 195: correct i.e, to i.e.,

**Quality:**

3

**Strengths And Weaknesses:**

## Strengths
1. The score-induced Birkhoff decomposition is a novel construction that selects a permutation order based on a score matrix, effectively resolving the non-uniqueness of the decomposition. This design enables differentiable extensions of discrete objectives and facilitates gradient-based optimization over permutations.

2. The theoretical analysis is solid, covering differentiability, minima correspondence, and a rounding guarantee that ensures relaxed solutions can be safely mapped back to valid permutations without degrading objective quality.

3. The proposed Frank-Wolfe algorithm is general and versatile, applicable to both neural combinatorial optimization and classical matching-based problems. Its effectiveness is empirically validated through comprehensive experiments on QAP, TSP, and DFASP tasks.

## Weaknesses
1. The decomposition has $O(n^5)$ computational complexity, and while truncated in practice, the paper lacks an analysis of how such truncation affects solution accuracy or theoretical guarantees.

2. The experimental insights are limited. The key hyperparameters, such as truncation depth and score update frequency, are fixed without accompanying ablation studies or sensitivity analysis.

3. The advantage of using the Birkhoff decomposition over direct continuous optimization on the Birkhoff polytope (e.g., using Sinkhorn or projected gradient methods) is not clearly justified, and a more detailed discussion would be beneficial.

---

> ### Author Rebuttal · Authors · 2025-07-31
>
> We thank the reviewer for their constructive comments. We have performed a variety of ablation studies in response.
>
> Including evaluation of the dependence on truncation and score update frequency is a great suggestion. We have performed these experiments and will update the manuscript to include curves showing these dependencies. Our results show that, for $k \in \{3,5,10,15,20\}$ the greatest difference in performance is about 0.88\% in terms of avg. gap from optimal values. Furthermore, the difference from $k = 5$ to $k = 10$ (which achieves the best performance) is about 0.24\%. Note that we do not include the plots here due to restrictions from Neurips on the response format.
>
> | **Truncation size (k)** | **Gap from optimal % ↓** |
> |-------------------------|--------------------------|
> | $k=3$                   | 4.78\%                   |
> | $k=5$                   | 4.14\%                   |
> | $k=10$                  | 3.90\%                   |
> | $k=15$                  | 3.92\%                   |
> | $k=20$                  | 4.34\%                   |
>
> We also performed ablation experiments on score update frequency that we will amend the paper to include. In these experiments `update_score` is set to `True` every $m\in \{1,5,20,10,50\}$ epochs. These experiments show that the performance improves with higher frequency updates, that is, smaller $m$. Therefore, the best choice is always set `update_score = True`.
>
> | **Update frequency (m)** | **Gap from optimal % ↓** |
> |--------------------------|--------------------------|
> | $m=1$                    | 4.14\%                   |
> | $m=5$                    | 4.56\%                   |
> | $m=10$                   | 4.57\%                   |
> | $m=20$                   | 5.42\%                   |
> | $m=50$                   | 5.78\%                   |
>
> Despite the $O(n^3)$ complexity (with truncation), our method does scale to large instances. For example, for QAP, we test on QAPLIB which has an instances of size $n =256$. For this problem we get within 0.8\% of the best known solution using a runtime of 417 seconds.
>
> We agree that additional analysis on the truncated case would be beneficial. To this end, we will introduce a section formally introducing the truncated extension $F_S^{(k)} := \frac{1}{\sum_{i=1}^{k} \alpha_i} \sum_{i=1}^{k}\alpha_iP_i$. proving that all rounding properties  (Properties 3, 4-1, and 4-2) hold in the presence of truncation and renormalization of coefficients.
> This result follows easily since these properties only utilize characteristics of the first term in the decomposition and that extension is formed from a convex combination of discrete objective values. We will also add a proof of Theorem 2.11 in the truncating case. Again this is straightforward as the proof of this theorem only requires evaluating properties of the first term in the decomposition (ee duplicate comment to reviewer Qomh).
>
> The reviewer brings up a valuable point that there are many ways to optimize over the Birkhoff polytope: Frank--Wolfe, gradient descent + Sinkhorn, or, potentially, projected gradient descent. We performed experiments with both Frank--Wolfe and gradient descent + Sinkhorn, and found that Frank--Wolfe offered superior optimization results. In particular, using Frank-Wolfe yields an avg. gap from optimal of 4.14\%, whereas gradient descent + Sinkhorn gives a gap of 4.55\%. We will edit the manuscript to include these in the appendix.
>
> For each of these Birkhoff-polytope optimization methods, an objective over the polytope is required. Although there might be many options for this objective, we propose Birkhoff extension for its several advantageous properties, including differentiability and guaranteed rounding. For a comparison with an alternative objective, see the TSP experiments, where we evaluate optimization over a quadratic program relaxation. Here we demonstrate that superior results are achieved by initially optimizing the QP relaxation and then using this solution as the score matrix initialization. We invite further comments if there is a more direct objective we should consider.

---

> > ### Comment · Reviewer_s7f4 · 2025-08-03
> >
> > Thank you for the thorough response. Most of the concerns have been adequately addressed. Would it be possible for the authors to provide more detailed experimental settings and results for Q2 and Q4, including the associated runtime?

---

> > > ### Author Response · Authors · 2025-08-04
> > >
> > > Thanks for the question. Regarding the ablations, including the Sinkhorn experiments for Q4, we picked 14 problem instances from the QAPLIB with sizes from $n=12$ to $n=35$. We used a learning rate of $\eta = .001$ and time budget was always set to $2n$ seconds (identical to the experiments performed in the paper). Where not otherwise noted, we take $k = 5$. For the Sinkhorn + gradient decent experiments we used 10 iterations of Sinkhorn after every gradient decent step to achieve double stochasticity.
> > >
> > > Regarding Q2, we tested with a total runtime of $2n = 512$ seconds, but convergence to the final solution was achieved in $417$ seconds. This instance is part of QAPLIB and so the results we report from it are just what was achieved in the original experiments. Additional details on these experiments are available at line 351.
> > >
> > > If there are any other specifics you would like to see, please let us know.

---

> > > > ### Comment · Reviewer_s7f4 · 2025-08-05
> > > >
> > > > Thank you for the response. The previous concerns have been addressed. My only remaining concern is the trade-off between performance and runtime on large-scale tasks. If time permits, could the authors provide a more complete set of results—in tabular form—for instances of size 500 or 1000? It would be helpful to include comparisons with methods like Sinkhorn, reporting both performance and runtime.
> > > >
> > > > **I would be happy to increase my score if the method is validated on these more realistic problem sizes.**

---

> > > > > ### Author Response · Authors · 2025-08-07
> > > > >
> > > > > Thank you for your continued engagement.
> > > > >
> > > > > We have performed experiments on 5 QAP instances of size $n = 1000$ . Previous research on the QAP has been primarily limited to smaller instances. For example, QAPLIB, which is a compilation of many benchmarks, does not have any problem of size larger than $n =256$. (Note this instance is parameterized by two $256 \times 256$ matrices, and so is already quite large by some measures.) We therefore perform these new experiments on random instances. In particular, we randomly sample 1 000 facility points and 1000 location points independently and uniformly in \[0,1]², take the pairwise Euclidean distances among locations as the distance matrix $D$, and define the flow matrix $F$ to be inversely proportional to the pairwise Euclidean distances among the facility points.
> > > > > We hope that these experiments will serve as a proof of concept/scalability. The small number of instances is chosen due to time constraints, and we will perform the same tests on additional instances for the final version.
> > > > >
> > > > > The results are reported in the table below. As in the original manuscript we compare with Fast Approximate QAP (FAQ) and 2-opt (restart is a hyper parameter for this algorithm).  We evaluate both BE with Frank-Wolfe (FW) and BE with Sinkhorn projection (Grad + Sink). For the score matrix, we test with random initialization and initialization to the output of FAQ.
> > > > >
> > > > >
> > > > > | Method                               | Run-time (secs) | # of epochs | Avg. Objective ↓ | Gap from best % ↓ |
> > > > > | ------------------------------------ | --------------: | ----------: | ---------------: | ----------------: |
> > > > > | Random assignment                              |              –– |          –– |      124,016,604 |            59.41% |
> > > > > | 2-opt (Restart=3)                    |           2,000 |          –– |      103,795,243 |            33.48% |
> > > > > | 2-opt (Restart=5)                    |           2,000 |          –– |       87,551,100 |            12.53% |
> > > > > | FAQ                                  |             314 |          –– |       77,929,970 |             0.17% |
> > > > > | Ours (BE w/ Grad+Sink, Random Init.) |           2,000 |       9,826 |       90,257,464 |            16.00% |
> > > > > | Ours (BE w/ Grad+Sink, Random Init.) |           4,000 |      19,864 |       85,759,577 |            10.23% |
> > > > > | Ours (BE w/ FW, Random Init.)        |           2,000 |      10,030 |       88,592,034 |            13.87% |
> > > > > | Ours (BE w/ FW, Random Init.)        |           4,000 |      20,145 |       82,391,243 |             5.90% |
> > > > > | Ours (BE w/ FW, FAQ Init.)           |           2,000 |      10,176 |       77,800,750 |             0.00% |
> > > > >
> > > > > These results show that our method  produces reasonable results at instance sizes of $n = 1000$, despite runtime being restricted to $2n = 2000$  and $4n = 4000$ seconds. For these runtimes we are able to run around 10,000 and 20,000 epochs.

---

> > > > > > ### Comment · Reviewer_s7f4 · 2025-08-08
> > > > > >
> > > > > > Thank you for the authors' efforts. These new results are important for the community, as the scale of problems in deep learning [1] has already surpassed the sizes found in QAPLIB, which is why I am particularly interested in results on larger-scale tasks.
> > > > > > I have increased my score accordingly.
> > > > > >
> > > > > > [1]. Git re-basin: Merging models modulo permutation symmetries, ICLR, 2023.

---

### Official Review · Reviewer_Qomh · 2025-07-01

**Clarity:** 3
**Significance:** 4
**Originality:** 3
**Rating:** 5
**Confidence:** 4

**Summary:**

The paper proposes an approach for providing an differentiable extension to loss functions on permutation matrices using a novel decomposition of the Birkhoff polytope based on a total ordering of permutations created by the inner product to a particular 'score matrix'. The extension is differentiability almost everywhere since the decomposition is unique. Nonetheless, the extension may not be non-convex and the score matrix can be changed during operation to escape local minima. Additionally the extension has desirable rounding properties such that the permutation matrix obtained from the rounding operation to obtain the best permutation matrix within the decomposition is no worse than the extension's loss. Additionally if the score function is close to permutation than the rounding will only do better.

**Questions:**

The definition of $\mathcal{P}(B_k)$ could be made clearer. I understand it as the set of all permutations that are non-negative components of $B_k$, but note that $B_k$ itself is not double stochastic. Line 219 discusses maximum weight matching problem for the Birkhoff, but the residual $B_k$ isn't in the Birkhoff polytope. This should be clarified.

In Algorithm 3 and 4,  there could be a discussion of the necessitate of starting from a *dense* random double stochastic matrix $A$ (which should really be denoted $A_1$) compared to either the dense barycenter (a uniform double stochastic matrix) or a sparse combination. Especially since only 5 terms are initially used. Clearly there will be issues if the initial $A_1$ is not dense. But does it need to be random? Especially since the score matrix is is used the barycenter seems a logic initialization.

I assume that automatic gradient are necessary for the Frank-Wolfe algorithm as the relationship between extension and A through $\alpha_i$ is non-trivial to express.

Minor:
Line 319 minor typo "simple constraint" -> "simple constraints"

**Ethical Concerns:**

["NO or VERY MINOR ethics concerns only"]

**Final Justification:**

The paper provides a novel approach for an interesting combinatoric problems. The answers further clarified some minor points. There is some limitation for larger problems expressed by other reviewers. But this is a useful contribution. Strong accept.

**Limitations:**

Limitations are mentioned, but more discussion could be made.

**Quality:**

4

**Strengths And Weaknesses:**

Strength:

The paper contributes to combinatorial optimization through the use of differentiable extension.


Weakness:

There a few times things could be made clearer as noted in Questions below.

The practical implications of the truncation to 5 terms in the Birkhoff decomposition are not addressed by any of the theoretical results. This creates a gap between the theory and practice that deserves more discussion.

---

> ### Author Rebuttal · Authors · 2025-07-31
>
> We appreciate the thoughtful review and now respond to points raised.
>
> We agree that our results could be strengthened by closing the gap between theory and practice created by truncation. To this end, we will introduce a section formally introducing the truncated extension $F_S^{(k)} := \frac{1}{\sum_{i=1}^{k} \alpha_i} \sum_{i=1}^{k}\alpha_iP_i$, proving that all rounding properties  (Properties 3, 4-1, and 4-2) hold in the presence of truncation.
> This result follows easily since these properties only utilize characteristics of the first term in the decomposition and that extension is formed from a convex combination of discrete objective values. We will also add a proof of Theorem 2.11 in the truncating case. Again this is straightforward as the proof of this theorem only requires evaluating properties of the first term in the decomposition (see duplicate comment to reviewer S7f4).
>
> Thank you for suggesting the need for additional clarity with regards to $\mathcal P$. We define (Definition 2.2) $\mathcal P$ only as an operator on matrices $X$ with non-negative entries. For such matrices $\mathcal P(X)$ is the set of all permutations $P$ such that $X(i,j) >0$ if $P(i,j) = 1.$ This set is different than the set of permutations appearing in the Birkhoff decomposition; it contains all permutations that could potentially appear in a Birkhoff decomposition of $X.$  Line 219 discusses finding the permutation in $\mathcal P(X)$ that comes first in the total order. This operation does not require computing the decomposition, so there is not a remainder to address.
>
> Regarding the necessity of random initialization, since the extension is a.e. differentiable, random initialization ensures we initialize at a point that is differentiable (almost surely). This approach also has the advantage that optimizing from multiple initializations may yield superior results. However, it is very interesting to consider starting from a fixed doubly stochastic matrix. In this setting we can use multiple score matrix initializations to get multiple optimization trajectories. We performed experiments showing the effectiveness of initializing at the barycenter. We picked 14 problem instances from the QAPLIB with sizes from $n = 12$ to $n = 35$. We use identical hyperparameter (truncation depth $k = 5$, learning rate $\eta = .001$,  time budget $t = 2n$ seconds) and, for random and barycenter initialization, we evaluate the final average assignment costs across all 14 problem instances. These experiments show that random initialization version has avg. gap to optimal of 4.14\%, whereas the barycenter initialization has an avg. gap of 4.42\%. We will include these experiments in the manuscript.
>
> We do, indeed, use automatic gradients in practice and will include a comment stating this in the manuscript.  As the expression for the $k$th term in the decomposition includes $k$ nested min functions, the gradient is quite challenging to express.

---

> > ### Comment · Reviewer_Qomh · 2025-08-05
> >
> > Thank you for the sketch of truncated results.
> >
> > Thanks for the response regarding $\mathcal{P}(B_k)$, yes definition 2.2 is applicable to the residual $B_k$ I mentioned in Alg. 2 that is mentioned in Theorem 2.7. I understand that on line 219 there is no concept of a residual for the maximum weight matching.
> >
> > Thanks for the results for the barycenter. It seems logical and adds to the results. Can you confirm whether "we can use multiple score matrix initializations to get multiple optimization trajectories" is done? If so, is the same number of multiple random initializations performed? The difference in the gap seems small. Consider the interaction between the randomized score matrix and the initialization, it seems that randomized initialization would actually create a smaller set of initial truncated decompositions as compared to the barycenter.  That is if variance is taken across the random score matrices for a fixed random initialization versus the barycenter, the former would be lower.  However, as mentioned if random initializations are also taken than there may be more diverse initial decompositions.
> >
> > Thanks for clarifying the automatic gradient point.

---

> > > ### Author Response · Authors · 2025-08-07
> > >
> > > Thanks for the additional question, and sorry for any confusion we have introduced. We do not perform experiments with multiple trajectories either for the barycenter initialization or random initialization. This is certainly an interesting direction for future research given the interesting effect the score matrix has on the optimization. Indeed, it seems greater variance could be achieved by randomizing the score matrix and fixing initialization to barycenter (compared to random initialization).

---

### Official Review · Reviewer_6Krw · 2025-07-02

**Clarity:** 4
**Significance:** 3
**Originality:** 3
**Rating:** 5
**Confidence:** 4

**Summary:**

This paper presents a continuous and almost everywhere differentiable extension from functions over permutations to functions over doubly stochastic matrices. The starting point is the Birkhoff decomposition, which allows us to decompose any doubly stochastic matrix into a convex combination of a relatively small number of permutation matrices. If the doubly stochastic matrix $A$ is decomposed as $\sum_k \alpha_k P_k$, where the $P_k$ are permutation matrices, we might extend a function $f$ of permutation matrices to doubly stochastic matrices by defining $F(A) = \sum_k \alpha_k f(P_k)$. This would indeed be an extension, since when $A$ is a permutation matrix the only valid way to express it as a convex combination of permutation matrix is to put all the weight on $A$ itself. The main challenge with this approach is that the decomposition is not unique, and so depending on how we choose the decomposition to use for each doubly stochastic matrix $A$, the function $F$ may behave erratically. The core idea behind this paper is to introduce a total ordering over permutation matrices and to use that ordering to standardize the decomposition (by preferring to put weight on early permutation matrices in the ordering). The authors argue several nice properties hold with this modification: the extension $F$ is continuous and almost everywhere differentiable. Moreover, they define a collection of total orderings where the decomposition can also be computed in polynomial time (the orderings are those induced by inner products with scoring matrices, and as long as the scoring matrices have a bit of noise, this is guaranteed to be an ordering without ties). In this case, the decomposition of an $n \times n$ doubly stochastic matrix can be computed on $O(n^5)$ time. This means that the extension can be evaluated in $O(n^5)$ time to obtain the decomposition, and $O(n^2)$ calls of the function $f$ over permutations.

In addition to the above results, the authors also provide a number of other useful results: a version of the Frank-Wolfe optimizer that can be used to optimize $F$ without needing an expensive projection back onto the set of doubly stochastic matrices, tricks for changing the scoring matrix used to define the ordering to prevent a current optimization iterate from being at a local minima of the extended function, and extensions for functions of rooted binary trees.

Finally, the authors apply their extension to several combinatorial optimization problems (the quadratic assignment problem, traveling salesperson problem, and directed feedback arc set problem). Surprisingly, the extension finds better solutions than Gurobi and heuristic methods for QAP and DFASP under some regimes.

**Questions:**

Please see high level questions in strengths and weaknesses.

**Ethical Concerns:**

["NO or VERY MINOR ethics concerns only"]

**Final Justification:**

The authors adequately addressed my comments and after reading the other reviews I do not have any concerns with the paper so I will maintain my positive score.

**Limitations:**

Yes.

**Paper Formatting Concerns:**

None.

**Quality:**

4

**Strengths And Weaknesses:**

# Strengths
- The paper is very well written and despite being about a topic that I have relatively little background in, I was able to understand the main claims and arguments of the paper.
- Deep learning has shown that gradient based optimization can often work well even when we cannot prove that it should (e.g., when run on non-convex functions with many local minima). This paper provides a framework allowing the application of that machinery to combinatorial optimization problems over permutations, and the experiments show some evidence that this approach can in some cases outperform state-of-the-art solvers.
- The authors go well beyond establishing the basic idea and provide nice ideas for the practical application of their extension (e.g., updating the score matrix used to define the total ordering to escape local minima during optimization).

# Weaknesses
- The computational cost of computing the extension once is polynomial, but still relatively expensive ($O(n^5)$ time to compute the decomposition and $O(n^2)$ evaluations of the original function on permutations). In the experiments the authors only compute the first $k = 5$ coefficients in the decomposition. I am curious how much this cost can be reduced. I also wonder if there is any way to "warm-start" the decomposition so that if $A'$ is close to $A$ (i.e., maybe one step away in the gradient-based optimization), we can leverage the decomposition of $A$ to more efficiently decompose $A'$.
- It seems like the choice of scoring matrix $S$ might affect some properties of the extension (e.g., is the Lipschitz constant of the extension dependent on the score $S$?). In addition to the paper's results on escaping local minima, it could be interesting to explore whether the choice of $S$ has other impacts that could be optimized for.

---

> ### Author Rebuttal · Authors · 2025-07-30
>
> We are thankful for the especially thorough review, and glad the reviewer has a strong understanding of our work. We now address the questions raised.
>
> The reviewer's suggestion of warm starting the decomposition for $X$ using the previous GD iterate $X'$ is very interesting. In an extreme case, e.g. if $X'$ is a permutation matrix, the previous iterate's decomposition may contain very little information about the current decomposition (as the previous decomposition only has a single element). Therefore it seems challenging to reduce the worst-case complexity from $O(n^3)$ using this approach. On the other hand, in most cases the decompositions of $X$ and $X'$ should be very similar, so using a warm-start could offer a substantial practical speedup. This is an encouraging direction for future improvements. We have explored this observation to some degree in our implementation, where we have made the engineering choice to use a hash table that saves the results of common maximum weight matching problems. Using this technique, any maximum weight matching problems solved for computing the decomposition in the previous iterate will not be recomputed if these problems arise in computing the new decomposition.
>
>  It is compelling to consider, as the reviewer suggests, how the properties of the extension $F_S$ are dependent on $S$. We note a bound on the Lipschitz constant of the extension need not depend on $S$. The relevant quantity is instead $\Delta_f = \max_P f(P) - \min_Pf(P)$. Roughly, this arises because if two doubly stochastic matrices $A$ and $A'$ are within an $L_1$ distance $\epsilon$, then the $L_1$ difference in the weights  $\alpha_i$ of their decompositions is $O(\epsilon)$ (note that each coefficient is a 1-Lipschitz function of the doubly stochastic matrix). Thus, in the worst case, $|F(A) - F(A')|$ is $O(\epsilon \Delta_f)$, where we have suppressed the dependence on $n$. We will add a rigorous proof to the manuscript showing a Lipschitz bound independent of $S$. There may be other interesting ways in which the extension is dependent on $S$. These are compelling avenues for future work.

---

> > ### Comment · Reviewer_6Krw · 2025-08-02
> > **Thanks!**
> >
> > Thanks for the response!

---

### Decision · Program_Chairs · 2025-09-17

**Decision:**

Accept (poster)

**Comment:**

This paper presents a continuous and almost everywhere differentiable extension from functions over permutations to functions over doubly stochastic matrices, which is based on the Birkhoff decomposition. The proposed method exhibits several appealing properties, such as minima correspondence and a rounding guarantee. The authors further develop a Frank-Wolfe optimizer with optional dynamic score updates to help escape local minima, and they apply this framework to unsupervised neural combinatorial optimization. Experimental results on QAP, TSP, and DFASP tasks demonstrate the effectiveness of the proposed approach.

All reviewers agree that the paper makes an interesting and novel theoretical contribution on an important problem. Moreover, the experimental results are promising and clearly demonstrate the potential of the theoretical findings and the proposed method. All potential weaknesses identified by the reviewers (in particular, the high polynomial running time and the truncation in practice, which creates an inconsistency between the theoretical results and the experiments) have been addressed convincingly by the authors. Overall, after the discussion, no significant weaknesses remain.